# Serotonergic neurons regulate the *Drosophila* vascular niche to control immune stress hematopoiesis

Xiaohui Liu, Marianne Montemurro [ID] , Nathalie Vanzo [ID] [✉] & Michèle Crozatier [ID] [✉]

In adult mammals, hematopoietic stem/progenitor cells reside in the bone marrow, in a specialized microenvironment called a "niche", which is composed of different cell types, including nerves. Although it is established that sympathetic nerves regulate hematopoiesis, little is known about the role of neural serotonin in bone marrow. The *Drosophila* hematopoietic organ, the lymph gland, is aligned along the aorta, which corresponds to the vascular niche. Here, we report that serotonin signaling in the vascular niche regulates the hematopoietic response to an immune challenge. The serotonin receptor 1B expressed in vascular niche cells, together with serotonin produced by neurons regulate the degradation of the extracellular matrix of the lymph gland and prevent its premature dispersal after an immune challenge. Serotonin signaling in aorta cells acts via JAK/STAT pathway activation. Our results provide novel insights into how vascular niche cells integrate neural information to regulate lymph gland immune stress hematopoiesis.

In adult mammals, blood cell production is sustained throughout life by hematopoietic stem and progenitor cells (HSPCs) that reside in the bone marrow in a specific microenvironment called the "niche"[1–3]. Together, the endosteal (osteoblastic) and vascular/perivascular niches control HSPC properties. While our knowledge of HSPC/niche cell communication under normal conditions progresses, the processes that occur following an immune challenge remain more elusive[1–4]. Recent data indicate that the bone marrow hematopoietic niche is innervated by neurons of the parasympathetic system, but the regulatory role of this innervation has so far been very little studied[5–8].

Striking parallels exist between *Drosophila* and mammalian hematopoiesis. Immune blood cells in insects, called hemocytes, are related to the mammalian myeloid lineage[9], and two cell types, the plasmatocytes and the crystal cells, are involved in phagocytosis and melanisation/wound healing, respectively[10]. During larval stage, most blood cells are formed in the lymph gland (LG), a multi-lobed organ, which develops along the aorta[11–14]. The aorta is the anterior part of the cardiac tube (CT), the *Drosophila* vascular system. Under normal conditions third instar larvae lymph gland anterior lobes are divided into three zones: a medullary zone (MZ) containing a heterogenous population of progenitors (prohemocytes), a cortical zone (CZ) composed of differentiated plasmatocytes and crystal cells, and a non-hematopoietic group of cells called the posterior signaling center (PSC) (Fig. 1C)[11].

Two separate niches, the PSC and the cardiac tube, control hematopoietic homeostasis (i.e., the balance between multipotent progenitors and differentiated blood cells) under normal conditions[15–17]. At the end of the third instar, during metamorphosis, the lymph gland disrupts, and all hemocytes are released into the hemolymph[18]. However, following an immune challenge occurring during larval life, such as wasp parasitism[19], lymph gland homeostasis is drastically modified and shifts to immune stress hematopoiesis.

Following wasp parasitism, lymph gland progenitors enter proliferation and subsequently massively differentiate into lamellocytes[12,13,20]. The lamellocyte is a specific immune cell type that mediates the encapsulation and killing of pathogens too large to be phagocytosed[16,21–23]. The differentiation of lamellocytes in the lymph gland culminates with its rupture and the release of all lamellocytes

Molecular, Cellular & Developmental Biology, UMR5077, CBI/Université Paul Sabatier, Toulouse, France. [✉]e-mail: nathalie.vanzo@univ-tlse3.fr; michele.crozatier-borde@univ-tlse3.fr

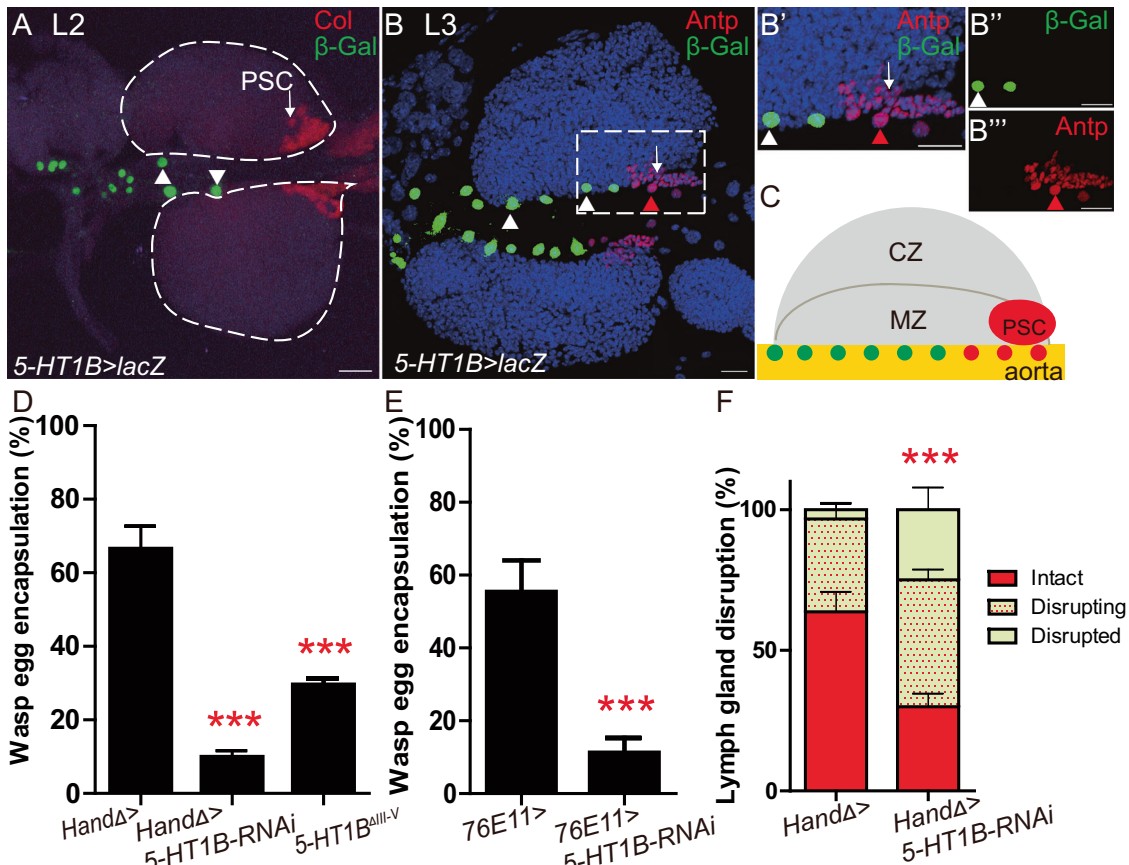

**Fig. 1 | 5-HT1B expressed in cardiac cells controls lymph gland dispersal and wasp egg encapsulation following wasp parasitism. A** *5-HT1B-Gal4(KI)>LacZ^{nls}* (green), and Col (red) expression in wild type (wt) second instar larvae (L2). *5-HT1B* (green) is expressed in a subset of anterior aorta cells (white arrowhead). **B**–**B'''** *5-HT1B-Gal4(KI)>LacZ^{nls}* (green), and Antp (red) expression in wild type (wt) third instar larvae (L3). *5-HT1B* (green) is expressed in all anterior aorta cells (white arrowhead) except those that express Antp (red arrowhead). **B'**–**B'''** Magnification of the white square drawn in (**B**) shows complementary stainings of LacZ (green, white arrowhead) and Antp (red, red arrowhead) in anterior aorta nuclei. **A**, **B** at least three independent experiments were analyzed. **C** Schematic representation of a third instar lymph gland anterior lobe, and anterior aorta (orange). Nuclei of cardiac cells expressing 5-HT1B are drawn in green while those expressing Antp are

in red. MZ and CZ represent medullary and cortical zones, respectively. **D**, **E** Quantification (%) of wasp egg encapsulation when *HandΔ>* (**D**) or *76E11* (**E**) are used as cardiac cell drivers to express *5-HT1B-RNAi*, or in a *5-HT1B* amorphic mutant (*5-HT1B^{ΔIII-V}*). The mean of three independent experiments is represented: control $n = 119$ (**D**) and 179 (**E**); *5-HT1B-RNAi* $n = 76$ (**D**) and 120 (**E**); *5-HT1B^{ΔIII-V}* $n = 120$ (**D**). **F** Quantification of lymph gland disruption (%) 10H post parasitism. Lymph glands were classified into three groups (see "Materials and methods"). Quantification represents the mean of three independent biological replicates: control $n = 85$, *5-HT1B-RNAi* $n = 76$. Error bars represent SEM. Statistical test: Pearson's Chi-squared test, one-tailed. *** $p < 0.001$. For (**D**–**F**), source data are provided as a Source Data file. Nuclei are labelled with DAPI (blue). Scale bars, 20 μm.

into the larval hemolymph, where they ensure wasp egg encapsulation. The role of the PSC in response to wasp parasitism is well documented[16,24–28], while the contribution of vascular niche cells has not been established so far.

Several studies report that neurotransmitters regulate lymph gland hematopoiesis. These neurotransmitters, produced in the brain, are released into the hemolymph and act directly on lymph gland cells[29,30]. The neurotransmitter serotonin (5-HT) is a monoamine produced by the decarboxylation of the tryptophan amino acid. Serotonin functions via specific serotonin receptors, and seven subtypes of serotonin receptors (5-HT1-7) have been identified in vertebrates (for review[31]). All the 5-HT receptors are seven-pass transmembrane G protein-coupled receptors (GPCRs), except 5-HT3, which is a gated ion channel receptor.

Serotonin and G protein-coupled serotonin receptors have a wide range of functions in modulating physiological and behavioral processes in animals. In addition, the serotonergic system has diverse non-neuronal functions, such as in embryonic development, pulmonary, vascular, cardiac, gastrointestinal, and reproductive functions (for review[32]). Serotonin also regulates multiple biological functions in the

mammalian hematopoietic system, such as embryonic development of HSPCs, differentiation and survival of hematopoietic cells, maintenance of the vascular system, and many hematological diseases are related with its dysregulation[33–35].

Serotonin and its GPCR receptors have been identified in many organisms, and *Drosophila* expresses five serotonin receptors: 5-HT1A and −1B, 5-HT2A and −2B, and 5-HT7[36–40]. The serotonergic system regulates key functions in the nervous system, such as sleep, circadian rhythm, feeding, aggression, nociception, and long-term memory formation[41,42]. Recently, a role for serotonin signaling in axis extension of *Drosophila* embryos has been reported[43]. Finally, serotonin modulates the phagocytosis by insect hemocytes[44]. Whether serotonin signaling contributes to *Drosophila* hematopoiesis has never been investigated.

Here, we provide evidence that in response to wasp parasitism, serotonin signaling is required in vascular niche cells and regulates, in a non-cell-autonomous manner, the extracellular matrix (ECM) meshwork of the lymph gland, thereby controlling the timely organ rupture, the release of lamellocytes, and ultimately the successful encapsulation of wasp eggs. We show that serotonin is produced by neurons, and

we establish a functional link between serotonin signaling and JAK/STAT pathway activation in vascular niche cells to control lymph gland dispersal and successful wasp egg neutralization.

## Results

### The serotonin receptor in the aorta regulates the anti-parasite response

The cardiac tube is composed of the aorta containing non-contractile cardiomyocytes, and the posterior heart chamber, where cardiomyocytes' rhythmic contractions provide the propelling force for blood flow/hemolymph circulation within the cardiac lumen[45]. The aorta is further subdivided into the anterior and posterior aorta, and the lymph gland is in contact with the anterior aorta (Fig. 1C)[46]. Single-cell RNA-sequencing data from our lab indicate that the serotonin receptor 1B (5-HT1B) is expressed in larval aorta cells (Crozatier et al., in preparation).

Interestingly, we observed that only anterior aorta cells express the 5-HT1B receptor, as reported by the 5-HT1B knock-in Gal4 allele (*5-HT1B-Gal4 (KI)*)known to reproduce endogenous 5-HT1B expression in the nervous system[47] (Fig. 1A, B). The same result is obtained with an independent 5-HT1B knock-in Gal4 allele[48]. In second instar (L2) *5-HT1B-Gal4 (KI) > UAS-LacZ^{nls}* larvae, β-Gal staining was detected at high levels in the nucleus of a few cells of the anterior aorta, whereas it was barely detected in other cardiac cells (Fig. 1A). In contrast to second instar larvae, all cells of the anterior aorta were labelled in third instar larvae (L3) (Fig. 1B), suggesting that expression begins at the second larval instar and extends to other cells of the anterior aorta in the third instar. Double staining with Antennapedia (Antp) which labels, the PSC and its neighboring anterior aorta cells[17], further indicated that aorta cells displaying high levels of Antp do not express 5-HT1B (Fig. 1B, C). Other serotonin receptors are not expressed in lymph glands or cardiac cells, with the exception of 5-HT7, which is expressed at very low levels in all cardiac cells (Supplementary Fig. 1A–C). Overall, these data reveal that 5-HT1B is expressed in anterior aorta cells lying in between the anterior lobes of the lymph gland.

Anterior aorta cells correspond to the vascular niche, known to control lymph gland homeostasis under physiological conditions[15]. We thus investigated whether aorta expression of 5-HT1B regulates lymph gland hematopoiesis under homeostatic conditions. From the second instar larval stage onwards, we inactivated *5-HT1B* expression in the cardiac tube, using the *HandΔ-Gal4* driver[15] and UAS-RNAi lines targeting 5-HT1B (*CT > 5-HT1B-KD*). We observed no difference in crystal cell, plasmatocyte, and core progenitor indexes in *CT > 5HT1B-KD* lymph glands as compared to the control indicating that under these normal conditions, 5-HT1B expression in aorta cells is not required for lymph gland homeostasis (Supplementary Fig. 1L–T).

However, it is not yet known whether the vascular niche regulates immune stress hematopoiesis induced by an immune challenge such as wasp parasitism. Following parasitic infestation, wasp egg encapsulation and lymph gland disruption reliably reflect the success of the *Drosophila* cellular immune response[28,49,50]. We therefore investigated whether in aorta cells, 5-HT1B plays a role to fight wasp parasitism. Wasp egg encapsulation was impaired in *CT > 5-HT1B-KD* as compared to control larvae (Fig. 1D). Similar results were obtained with a second *5-HT1B-RNAi* line (Supplementary Fig. 1E), or in an amorph *5-HT1B* mutant (Fig. 1D).

To determine whether 5-HT1B function is specifically required in the anterior aorta cells lying in between lymph gland anterior lobes, we knocked down *5-HT1B* using the *76E11-Gal4* driver restricted to anterior aorta cells of late second/third instar larvae[46]. *76E11-Gal4* driven *5-HT1B* knock down starting from the second instar stage onwards led to reduced wasp egg encapsulation as compared to the control (Fig. 1E and Supplementary Fig. 1F). This indicates that 5-HT1B is required in anterior aorta cells from the second instar larval stage onwards to allow successful wasp egg encapsulation. Similar results were obtained

with *GMR64A03-Gal4*, a distinct driver that is also expressed in anterior aorta cells (Supplementary Fig. 1D, G).

We next analyzed lymph gland disruption, 10H post-wasp egg laying. In control larvae (*HandΔ>*) 33% of the lymph glands were disrupting/disrupted, while in the *CT > 5HT1B-KD* larvae, this proportion increased up to 74% (Fig. 1F). Similar results were obtained with another *5-HT1B-RNAi* line (Supplementary Fig. 1H). We found that downregulation of 5-HT1B in cardiac cells using the HandΔ-Gal4 driver (*CT > 5-HT1B-KD*) does not affect heart lumen formation or heart rate (Supplementary Fig. 1I–K), ruling out any significant defect in the cardiac tube. Collectively, these data establish that in response to parasitism, 5-HT1B is required in anterior aorta cells to prevent premature lymph gland rupture and for successful wasp egg encapsulation.

### Defense against wasp parasitism requires neuronal serotonin

Serotonin receptors are activated by the binding of their ligand, serotonin. Previous studies have shown that nerve terminals are present in the aortic region of the heart[51,52]. The brain neurons innervating the aorta were called here Neurons[aorta]. However, no role for this aortic innervation has been reported to date.

We profiled serotonin expression using an antibody. To visualize Neurons[aorta] fibers we performed double immunostaining with Fasiclin 2 (Fas2) which labels axons from CNS neurons[53]. We observed low but reproducible expression of serotonin in nerve fibers and low levels of labelling in the cardiac tube (Fig. 2A–A"). Our data indicating that Neurons[aorta] fibers express serotonin is in agreement with a previous report[52]. We looked for Gal4 drivers expressed in Neurons[aorta] and identified four of them which are expressed in nerve fibers innervating the anterior aorta, in addition to CNS neurons, but not in cardiac cells (*Gyc89Da, GMR36B11, GMR13C09 and elav*) (Fig. 2B-B" and Supplementary Fig. 2A–C").

Serotonin is synthesized in a two-step process. First, in the rate limiting step, tryptophan hydroxylase (Trh) converts L-tryptophan to 5-hydroxytryptophan, which in the second step is cleaved by AADC/aromatic amino acid decarboxylase to yield serotonin. Trh is responsible for the production of neuronal serotonin[54]. The driver *Gyc89Da-Gal4* was used to knock down *Trh* using *Trh-RNAi (neuron>Trh-KD)*. A defect in wasp egg encapsulation (Fig. 2C) and a premature lymph gland disruption compared to the control were observed post parasitism (Fig. 2D). Similar defects were also observed using another *Trh* RNAi line (Fig. 2C and Supplementary Fig. 2E). Moreover, wasp egg encapsulation defects were similarly observed when any of the two *Trh* RNAi lines was driven by either of the other three Gal4 drivers expressed in Neurons[aorta] (Supplementary Fig. 2F–H), linking the production of serotonin by these neurons to the anti-parasite response.

We asked whether serotonin also regulates lymph gland hematopoiesis in the absence of parasitism. No difference in crystal cell, plasmatocyte and core progenitor indexes was observed in *neuron>Trh-KD* larvae compared to control (Supplementary Fig. 2I–Q), indicating that serotonin is not required to control lymph gland homeostasis under these conditions. These results are consistent with data observed in *CT > 5-HT1B-KD*. Collectively our data indicate that 5-HT1B expressed in anterior aorta cells and serotonin produced by neurons, possibly by those innervating the aorta, are required during parasitism to prevent premature lymph gland disruption and to allow an efficient wasp egg encapsulation.

### Neurons[aorta] establish synaptic contacts with anterior aorta cells

To visualize the spatial organization of Neurons[aorta] fibers along the aorta, we analyzed confocal cross-sections of the cardiac tube focusing at the level of anterior lymph gland lobes. We detected cardiac cells by the expression of membrane-tagged GFP (*HandΔ-Gal4 > mCD8-GFP*) and axons labelled with Fas2 antibody. We found that axons extend

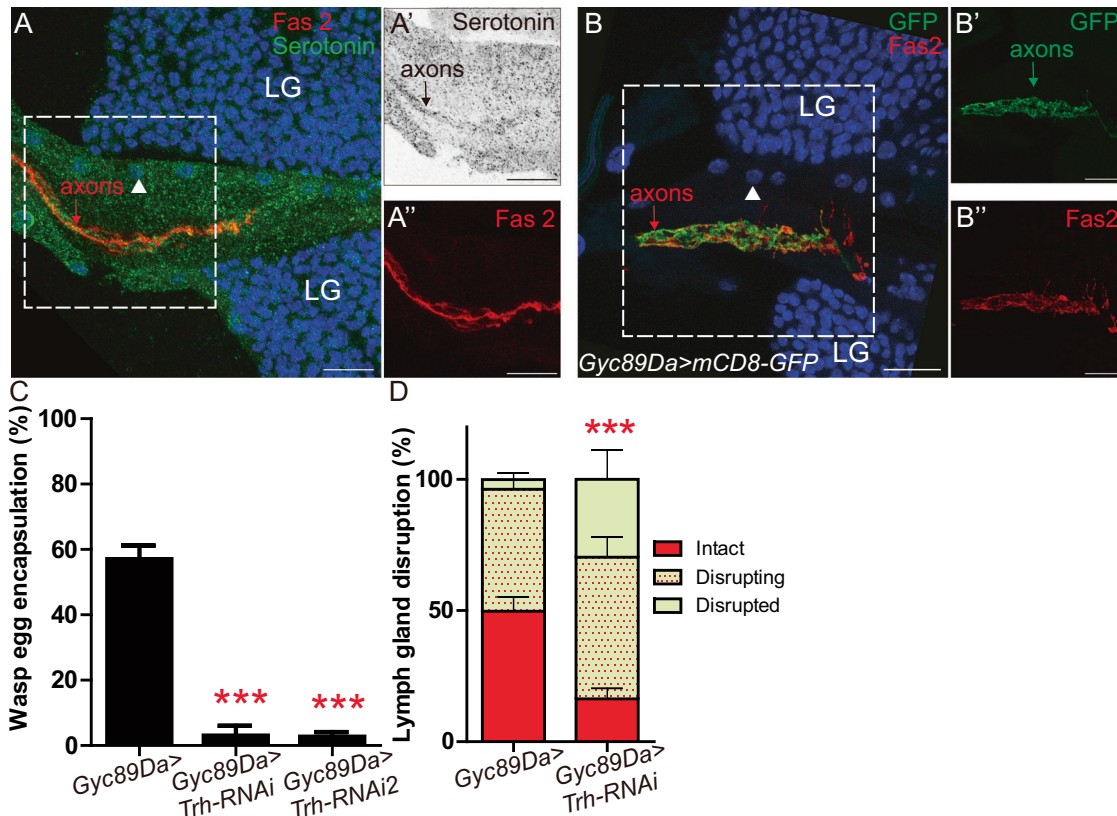

**Fig. 2 | Serotonin expressed in neurons is required for lymph gland rupture and wasp egg encapsulation following wasp parasitism. A–A"** Serotonin and Fas2 immunostaining in Neurons[aorta] axons in wt third instar larvae. Serotonin (green in **A**; black in **A'**) and Fas2 (red in **A** and **A"**). White arrowhead indicates anterior aorta nucleus. **B–B"** Fas2 (red) immunostaining of *Gyc89Da > mCD8GFP* (GFP, green) in Neurons[aorta] axons in wt third instar larvae. **A**, **B** At least three biological replicates were performed. **C** Quantification (%) of wasp egg encapsulation in larvae of indicated genotypes. The mean of three independent experiments is represented: control $n = 104$, *Trh-RNAi* $n = 71$, *Trh-RNAi2* $n = 74$. **D** Quantification of lymph gland disruption (%) 12H post parasitism in larvae of indicated genotypes. Quantification represents the mean of three independent biological replicates: control $n = 98$, *Trh-RNAi* $n = 84$. Error bars represent SEM. Statistical test: Pearson's Chi-squared test, one-tailed. *** $p < 0.001$. For (**C**, **D**), source data are provided as a Source Data file. Scale bars, 20 μm.

along the dorsal part of the cardiac tube and are in close proximity to aorta cells (Fig. 3A).

To further investigate cellular contacts between Neurons[aorta] axons and aorta cells, we carried out GRASP assays. GRASP uses complementary fragments of split-GFP (sp-GFP1-10 and sp-GFP11) expressed on the membranes of adjacent cells to reconstitute GFP fluorescence at contact points[55]. We identified that the *tow-LexA* driver is expressed in Neurons[aorta] (Supplementary Fig. 2D) and used it to express *LexAop-CD4-spGFP11, while UAS-CD4-spGFP1-10* was driven in cardiac cells using *HandΔ-Gal4*. Reconstituted GFP was observed along nerve fibers, establishing that these fibers are in close contact with aorta cells (Fig. 3B-B'). In the GRASP assay we performed Bruchpilot (Brp) immunostaining to label synapses[56,57]. We found that Brp colocalizes with reconstituted GFP all along the nerve fibers (Fig. 3B, B"), establishing that multiple synapses connect Neurons[aorta] fibers to aorta cells.

The Ca²⁺ dependence of neurotransmitter release is a fundamental property of chemical synapses[58,59]. Calcium-calmodulin-dependent proteinase kinase II (CamKII) is a key enzyme acting downstream of Ca²⁺. To interfere with Ca²⁺ signaling we knocked down CamKII in neurons using the *Gyc89Da-Gal4* driver (*neuron>CamKII-KD*)[46]. Following parasitism, we observed that wasp egg encapsulation was impaired in *neuron>CamKII-KD* larvae (Fig. 3C) together with premature disruption of lymph glands (Fig. 3D). Encapsulation defects were also observed using each of the three other neuronal drivers to knock down CamKII (Supplementary Fig. 3A–C).

Potassium channels are crucial regulators of neuronal excitability. We therefore inhibited neuronal activity by expressing the inward-rectifier potassium channel Irk2 (or Kir2)[60]. Defects in wasp egg encapsulation and premature lymph gland disruption were observed post parasitism (Fig. 3E, F). Collectively, these data indicate that neuronal activity is required to prevent premature lymph gland rupture and for wasp egg encapsulation following parasitism.

**Serotonin signaling regulates circulating lamellocyte numbers**

Upon parasitism, lymph gland hematopoiesis is associated with cell proliferation and differentiation into lamellocytes[28,61]. Previous data established that lamellocytes derived from lymph glands are essential for successful encapsulation of wasp eggs[28,62]. We thus investigated whether the defect in wasp egg encapsulation observed in *CT > 5-HT1B-KD* larvae could be linked to impaired lymph gland cell proliferation or lamellocyte differentiation. We investigated lymph gland cell division 6H post parasitism using phosphorylated histone H3 (H3P) immunostaining, and we found that the mitotic index in *CT > 5-HT1B-KD* was similar to the control (Fig. 4A–C), indicating that 5-HT1B is not required for lymph gland hemocyte proliferation following parasitism.

We then analyzed lamellocyte differentiation in the lymph gland in *CT > 5-HT1B-KD* and in *neuron>Trh-KD* using β-integrin immunostaining 6H post parasitism. Lamellocytes were present in lymph glands of *CT > 5-HT1B-KD* and *neuron>Trh-KD*, and lamellocyte indexes were similar to the control (Fig. 4D–I). Taken together, these data show that lamellocyte differentiation in lymph glands is not dependent on

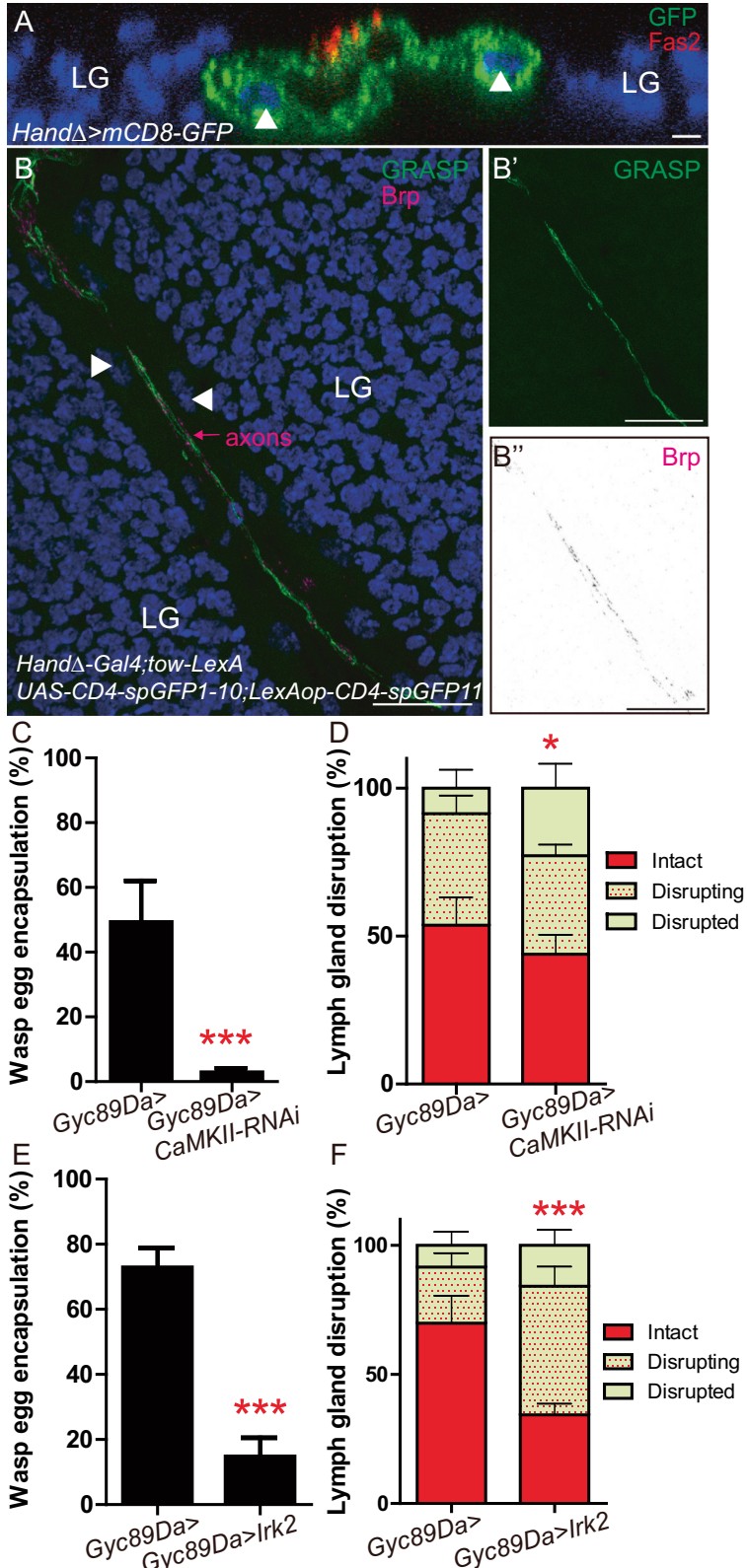

5-HT1B expression in aorta cells, nor on serotonin produced by neurons.

The ability of *Drosophila* larvae to encapsulate the wasp egg correlates with hemocyte numbers in the hemolymph, including lamellocytes[63]. Since lymph glands rupture prematurely in *CT > 5-HT1B-KD* parasitized larvae compared to the control, we analyzed the % of lamellocytes released at lymph gland rupture in these two genetic backgrounds. In control larvae lamellocytes correspond to 20% of circulating hemocytes when lymph glands disrupt at 14H post parasitism, while in the *CT > 5-HT1B-KD* larvae, this number decreases to 10% when lymph glands disrupt 12H post parasitism (Fig. 4J). Since lamellocyte differentiation in the lymph gland is similar in both contexts (Fig. 4D–I), this difference in circulation shows that lymph gland hemocytes undergo additional proliferation and/or differentiation to

**Fig. 3 | Synapses are present between Neurons^aorta and aorta cells and neuronal activity regulates lymph gland rupture and wasp egg encapsulation following wasp parasitism. A** Transversal section of anterior aorta. Cardiac cells express a membrane-bound GFP *(HandΔ>mCD8-GFP*, green) and Fas2 (red) labels axons. Nuclei of aorta cells are indicated by white arrowheads. Fas2 staining is localized at dorsal part of cardiac tube. Lymph gland nuclei are stained with DAPI. **B–B"** GFP reconstitution across synaptic partners (GRASP) shows physical connections between anterior aorta cells and Neurons^aorta fibers. *tow-LexA* and *HandΔ-Gal4* are used to express *LexAop-CD4-spGFP11* and *UAS-CD4-spGFP1-10*, respectively. GRASP is in green (**B, B'**), and Neurons^aorta synapses are stained with Brp immunostaining (pink in (**B**) or black in (**B"**)). Nuclei of aorta cells are indicated by white arrowheads; (**A, B**) at least three biological replicates were performed. **C, E** Quantification (%) of

wasp egg encapsulation when *CaMKII-RNAi* (**C**) and *Irk2* (**E**) are expressed in neurons using *Gyc89Da-Gal4* driver. The mean of three independent experiments is represented: control n = 85 (**C**) and 306 (**E**); *CaMKII-RNAi* n = 75 (**C**) and *Irk2* n = 231 (**E**). **D, F** Quantification (%) of lymph gland disruption post parasitism when *CaMKII-RNAi* (**D**) and *Irk2* (**F**) are expressed in neurons. Lymph gland disruptions were analyzed 11H post parasitism (**D**) and 17H post parasitism (**F**), since larvae were raised at 29 °C and 22 °C, respectively. Quantification represents the mean of three independent biological replicates: control n = 98 (**D**) and 68 (**F**); *CaMKII-RNAi* n = 83 (**D**) and *Irk2* n = 66 (**F**). Error bars represent SEM. Statistical test: Pearson's Chi-squared test, one-tailed. * p < 0.05, *** p < 0.001. For (**C–F**), source data are provided as a Source Data file. Scale bars, 20 μm.

substantially expand their population before lymph gland rupture in the control compared to *CT > 5-HT1B-KD* larvae.

A few hours later, 21 h after parasitism, the percentage of lamellocytes measured in the *5-HT1B-KD* hemolymph was approximately 3 times lower than in the control (Fig. 4K). Taken together, these data support the proposition that serotonin signaling in aorta cells is not required for lamellocyte differentiation per se, but prevents premature lymph gland dispersal, thus ensuring the appropriate loading of lamellocytes into the hemolymph, necessary for successful wasp egg encapsulation.

### Serotonin signaling prevents lymph gland extracellular matrix degradation

In the lymph gland, clusters of hemocytes are surrounded by a dense meshwork of extracellular matrix (ECM)[18,61,64,65]. The hematopoietic organ dispersal in response to wasp parasitism is associated with disruption of the basement membrane surrounding the lymph gland[28].

In control larvae and 6H post parasitism, immunostaining of two ECM proteins, namely heparan sulphate proteoglycan Perlecan/ Terribly Reduced Optic Lobe (Trol) (Fig. 5A, D, Supplementary Fig. 4D, J) and Laminin (Lan) (Supplementary Fig. 4A, G) shows that a dense meshwork of ECM surrounds hemocyte clusters. However, 6H post parasitism, in *CT > 5-HT1B-KD* or *neuron>Trh-KD* lymph glands, display a significant decrease in Trol (Fig. 5B, C, E, F) and Lan (Supplementary Fig. 4B, C) as compared to the control. In non-parasitized *CT > 5-HT1B-KD* or *neuron>Trh-KD* lymph glands, Trol and Lan staining are similar to the control, indicating that there is no major defect in lymph gland ECM under homeostatic conditions (Supplementary Fig. 4D–F, G–I, J–L). We conclude that under parasitism, serotonin signaling regulates the ECM meshwork that envelops the lymph gland, preventing its premature rupture.

Wasp parasitism triggers cell proliferation and rapid growth of the lymph gland. We reasoned that the ECM of the lymph gland must be reorganized to expand its size without bursting. Metalloproteinases are secreted proteinases involved in reorganization and degradation of ECM[66]. We investigated in vivo metalloproteinase activity by analyzing gelatinase enzymatic activity using an in situ zymography assay[67]. Gelatin conjugated to quenched fluorescein can be cleaved by metalloproteinases, leading in turn to fluorescence release. In control lymph glands, fluorescence is barely detected in the absence of parasitism, while it appears 6H post parasitism, indicating that parasitism leads to increased proteinase activity in lymph gland cells (Fig. 5G and Supplementary Fig. 4M, N). Interestingly, we found that in *CT > 5-HT1B-KD* parasitized larvae, lymph gland fluorescence is significantly higher than in the control, indicating increased metalloproteinase activity (Fig. 5G–I). This finding is consistent with the disorganization of lymph gland ECM and the premature lymph gland dispersal observed in *CT > 5-HT1B-KD* larvae.

One member of tissue inhibitors of metalloproteinase (Timp) is present in *Drosophila*[68,69]. Expression in cardiac cells of *timp* slightly delays lymph gland dispersal in the control (Supplementary Fig. 4O), whereas in a *CT > 5-HT1B-KD* context (*HandΔ>Timp > 5-HT1B-RNAi*) it

leads to a significant rescue of lymph gland dispersal compared to *CT > 5-HT1B-KD* alone (Fig. 5J). A slight improvement in Trol staining was also observed in *HandΔ>Timp > 5-HT1B-RNAi* larvae compared to *5-HT1B-KD* alone (Supplementary Fig. 4P–R). Collectively, these data confirm that metalloproteinase activity contributes to premature lymph gland dispersal observed in *CT > 5-HT1B-KD*. Taken together, these results suggest that under parasitism, serotonin signaling inhibits protease activity in lymph gland cells, thereby preventing ECM degradation and consequently precluding lymph gland premature disruption.

### JAK/STAT signaling acts downstream of 5-HT1B to fight wasp parasitism

Serotonin signaling activates the JAK/STAT pathway in rodent somatic muscles[70,71] and JAK/SAT signaling is known to play a key role in *Drosophila* defense against wasp parasitism[72–74]. We thus hypothesized that serotonin signaling could activate JAK/STAT signaling in aorta cells in response to parasitism.

We analyzed the expression of *10xStat92E-GFP*, a reporter of JAK/STAT activation[75]. 6H post parasitism, GFP expression was detected in anterior aorta cells, whereas without parasitism there was very low GFP. These data suggest that in aorta cells, the JAK/STAT pathway is activated in response to parasitism (Fig. 6A, B'). 6H post parasitism *10xStat92E-GFP* is also induced in the PSC and at the cortex of wild type lymph glands (Supplementary Fig. 5A, B'). Interestingly, 6H post parasitism, in *CT > 5-HT1B-KD* larvae *10xStat92E-GFP* expression is barely detected in aorta cells while it is expressed in the PSC, establishing that 5-HT1B is required for activation of the JAK/STAT pathway in aorta cells in response to parasitism (Fig. 6C, D).

Next, we investigated whether JAK/STAT signaling in cardiac cells was required for wasp egg encapsulation and lymph gland dispersal following parasitism. When the JAK/STAT pathway was inhibited by expression in cardiac cells of an RNAi targeting the JAK kinase *hopscotch* (*hop, HandΔ>hop-RNAi*)[76], wasp egg encapsulation was impaired, and lymph gland disruption increased as compared to the control (Fig. 6E, F). Since inhibition of *5-HT1B* or the JAK/STAT pathway in cardiac cells has similar effects on the response to parasitism, we wondered whether they could be functionally linked. Expression of a constitutively active form of *hop* (*hop^Tum*) in cardiac cells of *CT > 5-HT1B-KD* larvae (*HandΔ>5-HT1B-RNAi>Hop^Tum*) improved wasp egg encapsulation and lymph gland disruption compared to *CT > 5-HT1B-KD* alone (Fig. 6G, H, and Supplementary Fig. 5C).

We then investigated whether, after parasitism, activation of JAK/STAT signaling in cardiac cells prevents lymph gland rupture by regulating the ECM. When the JAK/STAT pathway was inactivated in cardiac cells by expression of *hop* RNAi (*HandΔ>hop-RNAi*), a significant reduction in Trol staining was observed compared to control (Fig. 6I–K). Furthermore, Trol staining was slightly stronger when premature lymph gland rupture was partially prevented by hop^Tum expression in *5-HT1B-KD* larvae (*HandΔ>5-HT1B-RNAi>Hop^Tum*) (Supplementary Fig. 5D–F). Taken together, these data show that in response to parasitism, 5-HT1B and JAK/STAT signaling are involved in the same process and that in aorta cells 5-HT1B functions, at least in

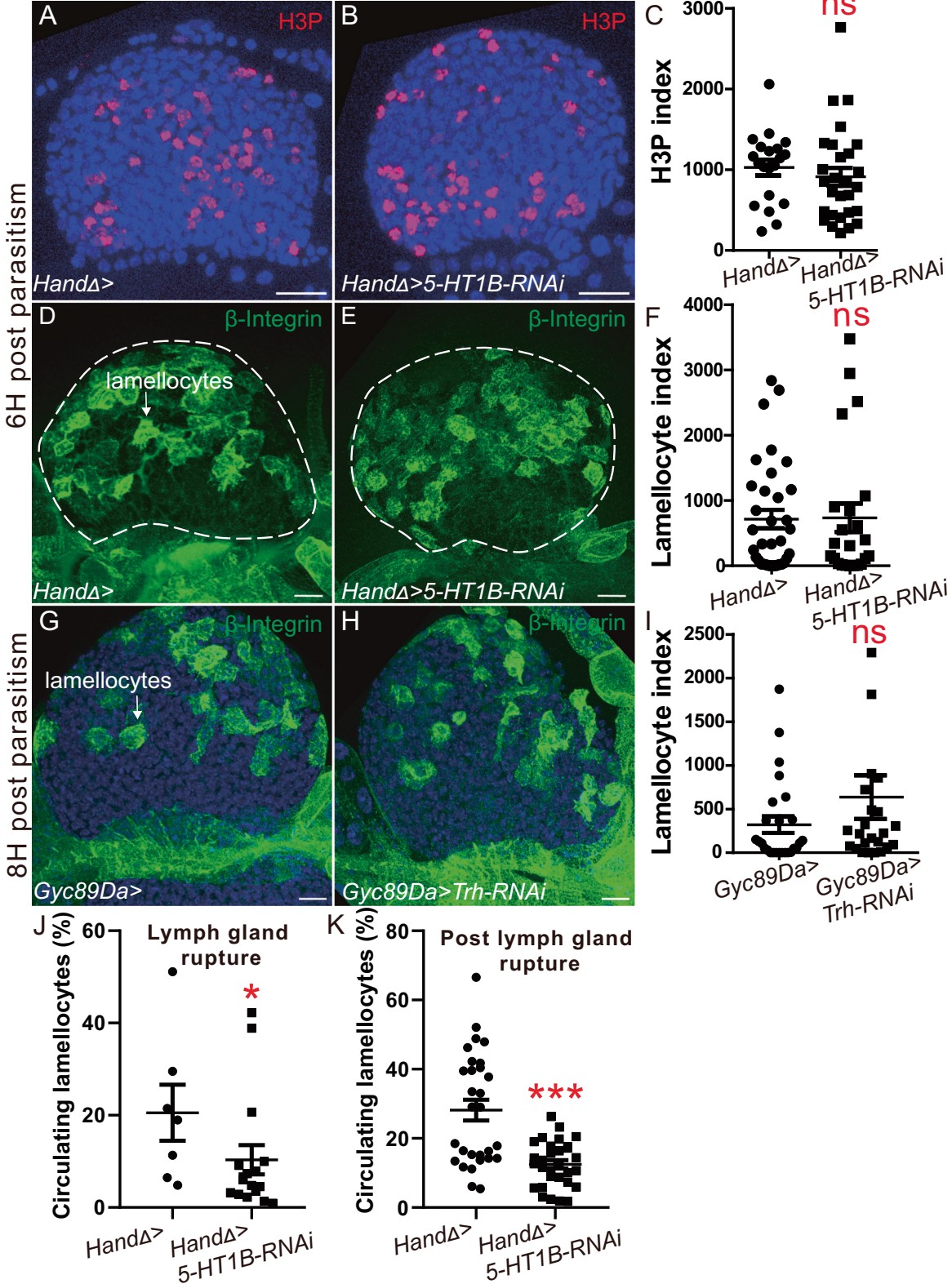

**Fig. 4 | Serotonin signaling is not required for lymph gland hematopoiesis in response to immune stress but ensures appropriate lamellocyte release into the hemolymph. A, B** H3P (red) detection 6H post parasitism in control (**A**) and in *HandD > 5-HT1B-RNAi* (**B**) lymph glands. **C** H3P index in lymph gland anterior lobes. **D, E, G, H** Lamellocyte differentiation (β-integrin, green) in control (**D, G**), *HandD > 5-HT1B-RNAi* (**E**), and *Gyc89Da>Trh-RNAi* lymph glands (**H**). **F, I** Lamellocyte index in lymph gland anterior lobes. **A–I** At least three independent experiments were performed, and one is shown. **J, K** Quantification (%) of circulating lamellocytes upon lymph gland rupture in control (14H post-parasitism) and *HandD > 5-HT1B-RNAi* (12H post-parasitism) (**J**) and 21H post parasitism (**K**). The mean of two (**J**) and three (**K**) independent experiments is represented: control *n* = 7 (**J**) and 29 (**K**); *5-HT1B-RNAi* *n* = 16 (**J**) and 30 (**K**). Error bars represent SEM. Statistical test: Mann-Whitney non-parametric test, two -tailed. * *p* < 0.05, *** *p* < 0.001, ns non-significant. For (**C, F, I–K**) source data are provided as a Source Data file. Scale bars, 20 μm.

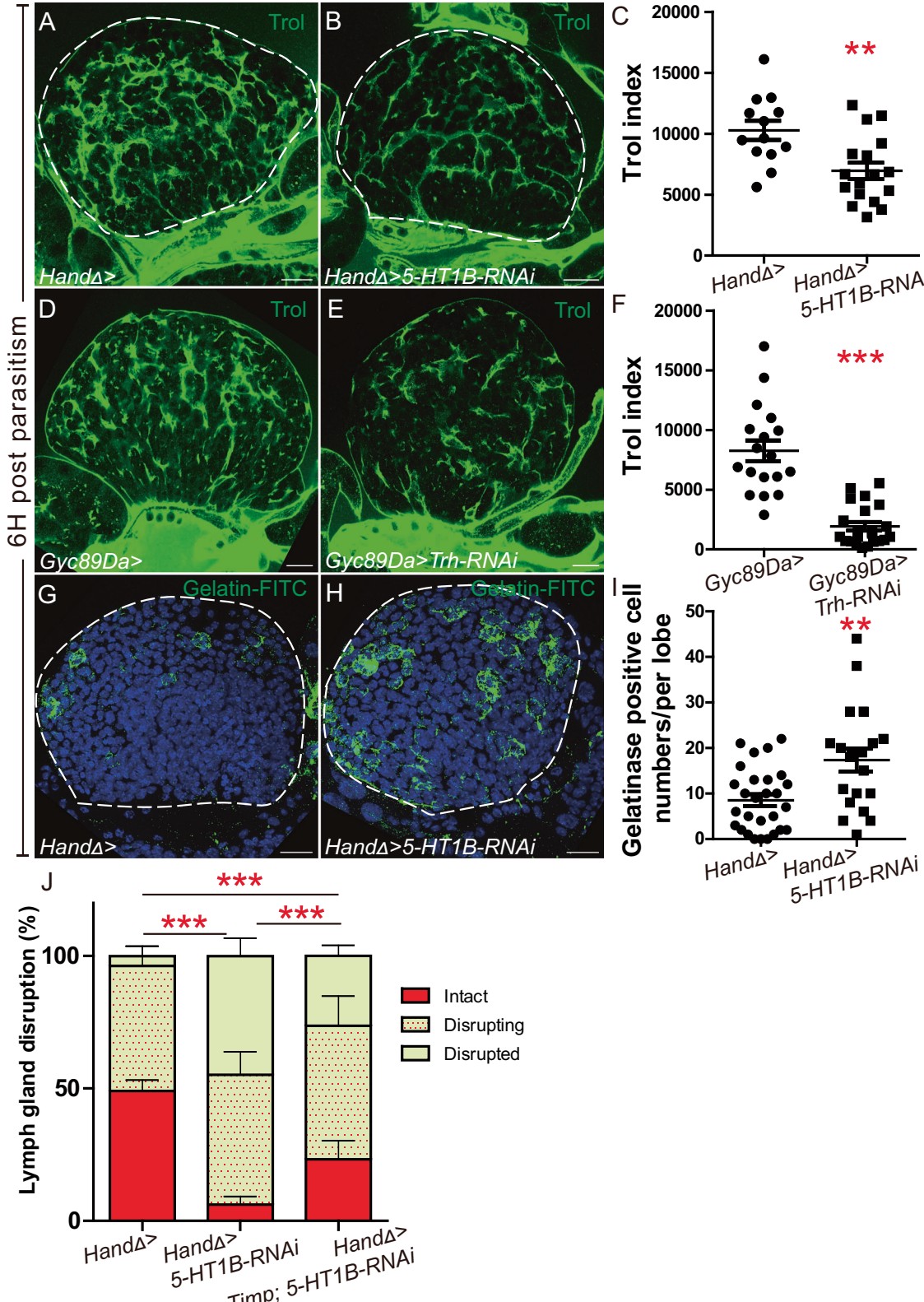

**Fig. 5 | Serotonin signaling prevents lymph gland extracellular matrix degradation following wasp parasitism. A**, **B**, **D**, **E** Trol immunostaining (green) 6H post parasitism in control (**A**, **D**), *HandΔ>5-HT1B-RNAi* (**B**), and *Gyc89Da>Trh-RNAi* (**E**) lymph glands. **C**, **F** Trol index in lymph gland anterior lobes. **G**, **H** Zymography assays 6H post parasitism using gelatin-FITC substrate (green) in control (**G**) and *HandΔ>5-HT1B-RNAi* (**H**). **I** Quantification of gelatinase positive cells per lymph gland anterior lobe. **A–I** At least three independent experiments were performed, and one is shown. **J** Quantification of lymph gland disruption (%) 11H post parasitism of indicated genotypes. Quantification represents the mean of three independent biological replicates: control *n* = 25; *5-HT1B-RNAi n* = 33 and Timp; *5-HT1B-RNAi n* = 34. Error bars represent SEM. Statistical test: Mann-Whitney nonparametric test, two tailed in (**C**, **F**, **I**) and Pearson's Chi-squared test, one tailed in (**J**). ** *p* < 0.01, *** p < 0.001. For (**C**, **F**, **I–J**) source data are provided as a Source Data file. Scale bars, 20 μm.

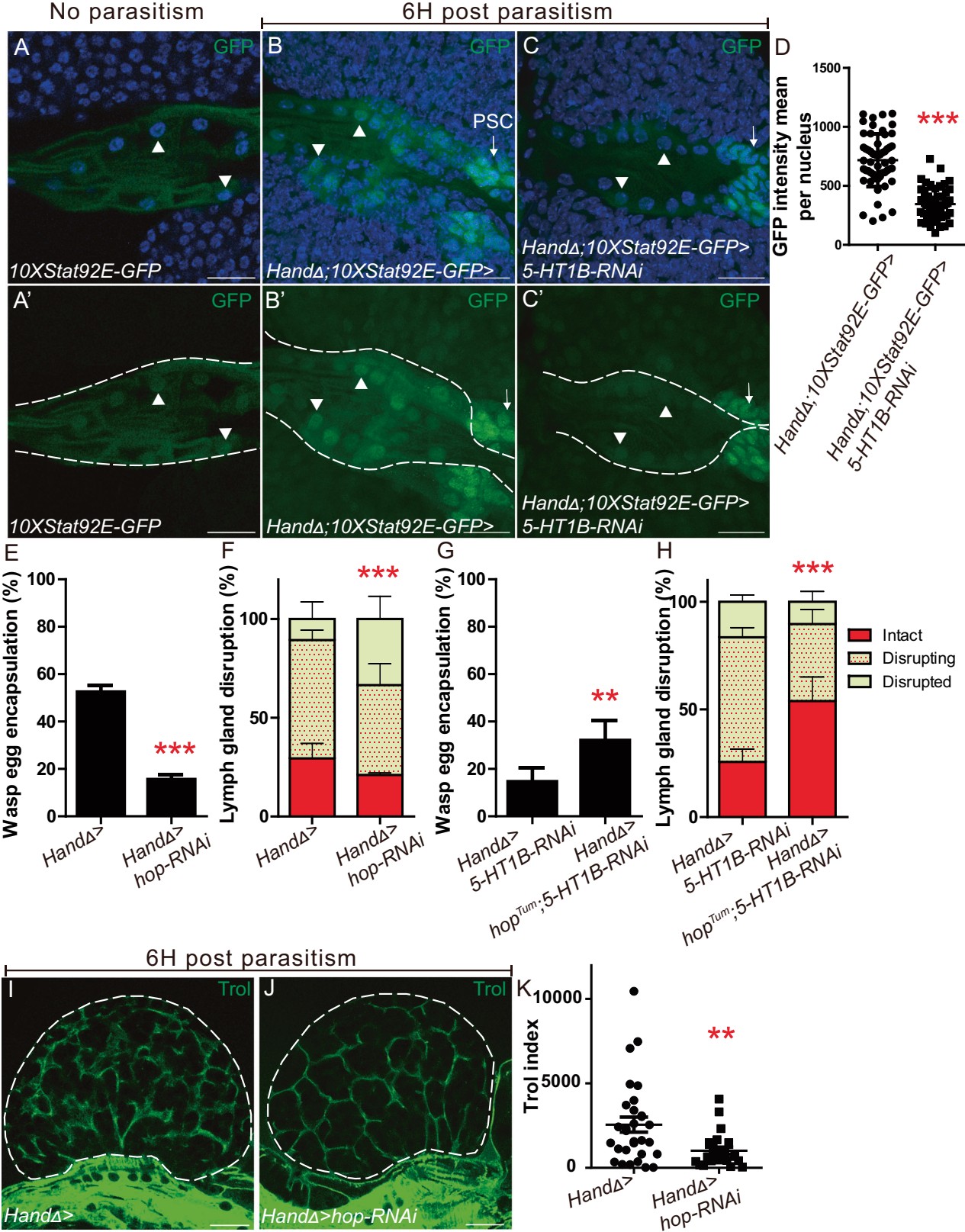

part, through JAK/STAT signaling to prevent lymph gland ECM degradation and premature dispersal of the lymph gland.

## Discussion

In this study, we demonstrate the existence of a dialogue between serotonin-producing neurons and vascular niche cells following wasp parasitism. Our results support the hypothesis that following an immune stress, serotonin produced by neurons binds the 5-HT1B receptor expressed in anterior aorta cells. Subsequent activation of JAK/STAT signaling in the vascular niche limits ECM degradation in lymph gland anterior lobes, thereby preventing the hematopoietic organ from disrupting prematurely (Fig. 7A). This, in turn, enables

**Fig. 6 | 5-HT1B activates JAK/STAT signaling in cardiac cells which controls lymph gland rupture and wasp egg encapsulation following parasitism. A–C′** Close up view of anterior aorta in control larvae expressing *10xStat92E-GFP* (green) without parasitism (**A–A′**) and 6H post parasitism (**B, B′**), and in *HandΔ>5-HT1B-RNAi* (**C–C′**) 6H post parasitism. (**A′–C′**) Dotted lines indicate cardiac tube outline. Cardiac cell nuclei and the PSC are indicated by white arrowheads and white arrows, respectively. Nuclei are labelled by DAPI (**A–C**). **D** Quantification of GFP intensity per cardiac cell nuclei. **A–D** At least three independent experiments were performed. **E, G** Quantification (%) of wasp egg encapsulation. The mean of three independent experiments is represented: control *n* = 218; *hop-RNAi n* = 233 (**E**);

*5-HT1B-RNAi n* = 152, *hop^Tum^;5-HT1B-RNAi n* = 132 (**G**). **F, H** Quantification of lymph gland disruption (%) 12H (**F**) and 8H (**H**) post parasitism. Quantification represents the mean of three independent biological replicates: control *n* = 71, *hop-RNAi n* = 56 (**F**); *5-HT1B-RNAi n* = 91, *hop^Tum^;5-HT1B-RNAi n* = 90 (**H**). **I, J** Trol immunostaining (green) 6H post parasitism in control *HandΔ>* (**I**) and *HandΔ>hop-RNAi* (**J**) lymph glands. **K** Trol index in lymph gland anterior lobes. Error bars represent SEM. Statistical test: Mann-Whitney nonparametric test, two-tailed in (**D, K**) and Pearson's Chi-squared test, one-tailed in (**E–H**). ** *p* < 0.01, *** *p* < 0.001. For (**D, E–H, K**) source data are provided as a Source Data file. Scale bars, 20 μm.

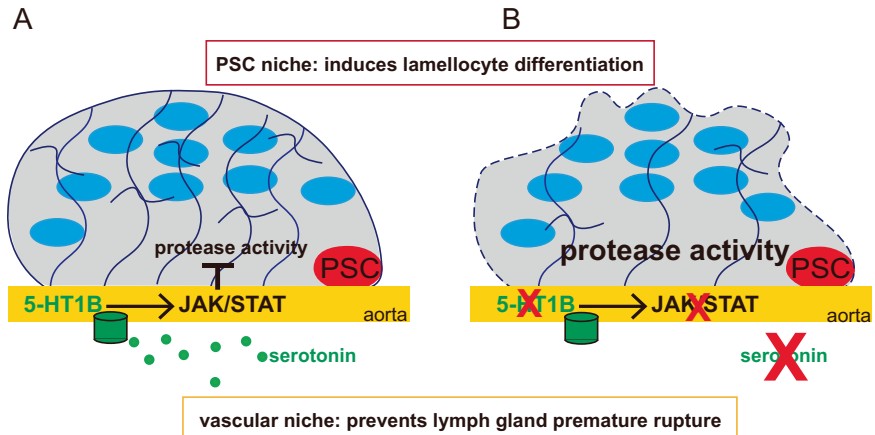

**Fig. 7 | Proposed model for the neuro/vascular regulation of lymph gland rupture upon wasp parasitism. A** Schematic representation of third instar larval lymph gland anterior lobe in control. Lamellocytes are drawn as blue ellipses, and the ECM is represented by a dark blue meshwork. Serotonin (green) released by neurons and/or Neurons^aorta, binds its receptor (green) expressed by anterior aorta cells and activates JAK/STAT signaling. This in turn non-cell-autonomously represses proteinase activity in lymph gland cells to delay ECM degradation,

thereby preventing the hematopoietic organ from rupturing prematurely. Hence, the lymph gland response to immune stress is controlled by two niches: the PSC, which is essential for lamellocyte differentiation, and the vascular niche, which via serotonin signaling prevents premature dispersal of the hematopoietic organ. **B** When serotonin signaling is defective, the lymph gland disrupts before all the lymph gland cells have differentiated, leading to a reduced load of lamellocytes into the circulation and a defect in the encapsulation of wasp eggs.

sufficient lamellocyte production and release into the circulation to ensure effective encapsulation of the wasp egg. In *CT > 5-HT1B-KD* larvae, premature dispersal of the lymph gland correlates with increased proteinase activity in lymph gland cells (Fig. 7B). The regulatory cascade established here is required for the defense against wasp parasitism.

Finally, this study establishes for the first time that the vascular niche is involved in the lymph gland immune response to parasitism (Fig. 7A). Hence, the lymph gland response to immune stress is controlled by two niches: the PSC, which is essential for lamellocyte differentiation via activation of the Toll/NF-κB and EGFR signaling pathways[27,28], and the vascular niche, which prevents premature dispersal of the hematopoietic organ via serotonin signaling (Fig. 7A). These two niches are both necessary, as they act independently on different aspects of the immune response to effectively combat wasp parasitism.

This study establishes that 5-HT1B from aorta cells regulates lymph gland immune stress hematopoiesis. Published single cell RNA sequencing data show that 5-HT1B is expressed in the PSC and in some circulating blood cells[77–80]. Rodrigues et al.[81] report that a 5-HT1B reporter is expressed in the posterior lobes and in MZ progenitors. By single cell transcriptomics and using two independent *5-HT1B-KI* transgenic lines, we further establish that 5-HT1B is expressed in anterior aorta cells. Taken together, these data reveal a complex expression profile of 5-HT1B in blood cells. It remains to be tested whether 5-HT1B expressed in other blood cell types also participates in hematopoiesis regulation.

In aorta cells, serotonin signaling activates the JAK/STAT pathway in a cell-autonomous manner. JAK/STAT signaling plays a positive and

complex role in *Drosophila* cellular immune defense. In the lymph gland, JAK/STAT signaling affects progenitor maintenance and the response to wasp parasitism[11,16]. Latran, which codes for an inactive receptor, binds to the Domeless receptor and forms a heterodimeric complex that prevents the activation of JAK/STAT signaling under homeostatic conditions. In response to parasitism, Latran expression is upregulated, which turns off JAK/STAT signaling and promotes the differentiation of progenitors into lamellocytes[72]. Moreover, in response to parasitism, the JAK/STAT signaling pathway is activated in larval somatic muscles and, by acting on carbohydrate metabolism, is necessary for lamellocyte differentiation and encapsulation of the wasp egg[74,82].

Interestingly, our study reveals a new non-cell-autonomous role for the JAK/STAT pathway in the control of lymph gland response to an immune stress. In vascular niche cells, JAK/STAT signaling is activated downstream of 5-HT1B to regulate lymph gland ECM to prevent lymph gland dispersal. How 5-HT1B activates JAK/STAT signaling is unclear, but Guillet-Deniau et al.[71] provide a possible explanation. Indeed, in rat myoblasts, binding of serotonin to its 5-HT2A receptor triggers rapid and transient tyrosine phosphorylation of JAK2 and activation of JAK/STAT signaling. The authors also report that the 5-HT2A receptor and STAT3 co-precipitate with JAK2, indicating that they are physically associated. In addition, another study reports that in rat vascular smooth muscle cells, serotonin activates JAK2, JAK1 and STAT1 via 5-HT2A receptor[70]. Whether 5-HT1B activates JAK/STAT signaling in aorta cells, independently of a ligand, as it does in rat muscles, is an interesting possibility that remains to be investigated.

In *Drosophila* aorta cells, JAK/STAT signaling downstream of 5-HT1B may prevent premature lymph gland rupture by delaying ECM

degradation, accompanying the lymph gland dispersal in response to wasp parasitism[28]. Matrix metalloproteinases (MMPs) are secreted proteinases, which play multiple roles both in ECM remodeling and protein turnover. Several studies establish functional links between JAK/STAT signaling and the activity of MMPs. The MMP superfamily includes the "classical" MMPs (MMP1 and MMP2), the ADAMS (a disintegrin and metalloprotease) and the ADAMTS (a disintegrin and metalloproteinase with thrombospondin motif)[66].

In mammalian osteoarthritis, a chronic and progressive age-related degenerative disease caused by excessive chondrocyte death and subsequent degradation of the ECM, JAK/STAT signaling is activated in the chondrocytes of the intervertebral disc and controls cartilage remodeling, by regulating the expression of MMPs[66,83]. In *Drosophila*, JAK/STAT signaling is involved at different steps during oogenesis, including controlling egg chamber shape[84-86]. Wittes et al. [87] establish that in terminal follicle cells, activation of the JAK/STAT pathway controls the expression of a secreted AdamTS-A, which remodels the basement membrane at the poles of the egg chamber to promote its elongation along the anterior-posterior axis. Understanding how 5-HT1B and JAK/STAT signaling in aorta cells might non-cell-autonomously regulate metalloproteinase activity in lymph gland cells needs further investigation.

In *Drosophila*, little is known about neuronal and immune cross-talk. In larvae, the sessile hemocytes present under the cuticle, known as hematopoietic pockets, are controlled by the peripheral nervous system. The neurons supply Activin-β/TGF-β (Transforming Growth Factor) to promote the adhesion and proliferation of hemocytes in the hematopoietic pockets[88,89]. Recent studies reported on neuronal regulation of lymph gland hematopoiesis under homeostatic conditions[29,30,90]. However, neuronal control of lymph gland hematopoiesis in response to immune stress remains largely unexplored.

Here, we provide evidence that serotonin signaling in vascular niche cells regulates lymph gland hematopoiesis in response to parasitism. Our working model is that in response to wasp parasitism, serotonergic neurons are activated and release serotonin. Serotonin binds to the 5-HT1B receptor expressed by anterior aorta cells, and serotonin signaling activates, at least in part, the JAK/SAT pathway in a cell-autonomous manner to prevent lymph gland rupture. How serotonergic neurons are activated by wasp parasitism is undetermined and little is known about how the wasp attack modifies *Drosophila* hematopoiesis.

Several studies reported on the role of a subset of neurons of the peripheral nervous system (PNS), the multiple dendritic arborization (da) neurons known as class IV (cIV da) neurons. cIV da neurons are mechanical nociceptors whose receptive territories cover the entire larval epidermis, and are involved in the response to parasitoid attack[91-96]. Upon wasp parasitism, cIV da neurons are required in the bending, rolling and crawling of larvae to escape wasp attack. The neural circuits acting downstream of cIV da neurons in this behavior are beginning to be deciphered[97-100]. Furthermore, cIV da neurons are involved in lamellocyte production in response to parasitism, but the neural circuit involved remains unknown.

Our study shows that a subset of serotonin-producing neurons in the brain prevents premature rupture of the lymph gland following parasitism. We can therefore suggest that there may be a link between cIV da neurons and serotonergic neurons regulating hematopoiesis in response to parasitism. One possibility is that the cIV da neurons detect the wasp attack and, via as yet unknown neural networks, activate the serotonergic neurons and serotonin release.

The serotonergic neurons involved have not been identified, nor have those that project towards the aorta. However, to inhibit serotonin synthesis, we used four independent neuronal drivers. They all have distinct expression patterns in brain neurons but share a common expression pattern in Neurons[aorta]. Since the same hematopoietic effect was observed with all four drivers, it is reasonable to propose that this effect was due to reduction of serotonin in Neurons[aorta]. Of course, it cannot at this point be totally ruled out that a common expression in other neurons is responsible. Challenging objective for the future is the identification of the neural circuits regulating lymph gland response to parasitism. It requires the identification of the neurons, among some 100 serotonergic neurons present in the larval brain, that are required here, to establish whether Neurons[aorta] are involved, and finally to explore any potential dialogue between serotonergic neurons and cIV da neurons.

In the mammalian bone marrow the nervous system regulates hematopoiesis by acting on HSPCs both cell-autonomously and non-cell autonomously through the niche[5,6,101,102]. Serotonin is involved in the development of HSPCs during vertebrate embryogenesis. While in zebrafish, CNS-derived serotonin increases the number of HSPCs emerging from the aorta-gonad-mesonephros (AGM)[103], in mice, serotonin synthesized in AGM endothelial cells promotes the survival of HSPCs in intra-aortic hematopoietic clusters, mainly by inhibiting the pro-apoptotic pathway[104,105]. In adult mammalian bone marrow, serotonin is linked to bone homeostasis and the differentiation of various hematopoietic lineages. In addition, HSPCs express serotonin receptors, and serotonin has been shown to exert a positive impact on the proliferation or differentiation of HSPCs in vitro. It has been suggested that in vivo, in the bone marrow serotonin may regulate the maintenance and regeneration of HSPCs after injury[6,33].

Numerous studies have established a close parallel between hematopoiesis in the lymph gland of *Drosophila* and that taking place in the bone marrow of mammals[106,107]. The activation of serotonin and JAK/STAT signaling in the vascular niche, and the subsequent modification of the ECM observed in lymph gland cells in response to an immune challenge, which we report in the present study, might therefore also be relevant to mammalian hematopoiesis. In the bone marrow, metalloproteinases are known to play a key role in the regulation of hematopoiesis by controlling ECM organization, protease-dependent niche factor availability and cellular interactions[108]. Whether in the bone marrow, serotonin signaling contributes to ECM remodeling following injury remains to be investigated and might provide a new entry point towards the elucidation of mechanisms involved.

## Methods
### Fly strains
$w^{1118}$ (wild type, *WT*), *UAS-mCD8GFP*[16], *HandΔ-gal4*[15], *76E11-Gal4* (BL39933[46]), *5-HT1A-Gal4 (KI)*, two independent lines for *5-HT1B-Gal4 (KI)*, *5-HT2B-Gal4 (KI)*, *5-HT7-Gal4 (KI)*[47,109], *10XStat92E--GFP*[75]. Other strains provided by Bloomington (BL) and Vienna (VDRC) Drosophila stock centers: *5-HT1B*[ΔIII-V] (BL55846), *UAS-Dicer* (BL24650), *UAS-LacZ*[nls] (BL3956), *UAS-GFP*[nls] (BL4775), *tub-Gal80*[ts] (BL7019), *UAS-Irk2, UAS-CaMKIIRNAi*[46], *GMR64A03-Gal4(CrebA-Gal4)* (BL46512), *UAS-5-HT1B-RNAi* (BL33418 and BL25833), *UAS-Trh-RNAi* (BL25842 and VDRC105414)*, GMR13C09* (BL48555)*, Gyc89Da* (BL93111), *GMR36B11-Gal4* (BL47560), *elav-Gal4*(gift of G. Isabel); *tow-LexA* (BL81564), *UAS-CD4spGFP1-10; LexAop-CD4-spGFP11* (BL58755), *UAS-Hop-RNAi*(BL32966[87])*, UAS-Hop*[Tum], *UAS-Timp* (BL58707), *LexAop2-mCD8-GFP* (BL32205).

Crosses and subsequent raising of larvae until late first/early second instar larval stage were performed at 25 °C, before shifting larvae to 29 °C for RNAi treatments until their dissection at the third instar larval stage. Controls correspond to Gal4 drivers with *UAS-Dicer* crossed with $w^{1118}$. For Gal80[ts] temperature shift experiments, crosses were initially maintained to 18 °C (permissive temperature) for 4 days after egg laying, and then shifted to 29 °C until dissection (Supplementary Fig. 1l–T). The same conditions were used for *Gyc89Da>Trh-RNAi* (Supplementary Fig. 2I–Q). For Irk2 experiments *Drosophila* laid eggs for 12H at 25 °C then were shifted at 18 °C for five and half. Days.

4H before wasp parasitism larvae were placed at 25 °C and then shifted to 22 °C until their dissection.

## Immunostaining

Lymph glands were dissected and processed as previously described[16]. Primary antibodies were mouse anti-Hnt (1/100, Hybridoma Bank), mouse anti-Col (1/40)[16], mouse anti-P1 (1/30, I. Ando, Institute of Genetics, Biological Research Center of the Hungarian Academy of Science, Szeged, Hungary), mouse anti-β-Intergrin (1/100, Developmental Studies Hybridoma Bank, CF.6G11c), chicken anti-β-gal (1:1000, Abcam), rabbit anti-Trol (1/1000) and rabbit anti-Laminin (1/750) gift from S. Baumgartner, mouse anti-Fas2 (1/20, DSHB), mouse anti-Brp (1/20, DSHB), rabbit anti-Serotonin (1/1000 Sigma-Aldrich), rabbit anti-H3P (1/200, Upstate Bioscience), mouse anti-Antp (1/100, DSHB), chicken anti-GFP (1/500, Abcam). Secondary antibodies were Alexa Fluor-488 and -555 conjugated antibodies (1/1000, Molecular Probes) and goat anti-Chicken Alexa Fluor-488 (1/800, Molecular Probes). Nuclei were labeled with DAPI.

## Wasp parasitism, lymph gland rupture, and wasp egg encapsulation

Late second instar or early third instar *Drosophila* larvae raised at 25 °C and shifted to 29 °C (for RNAi experiments) were subjected to parasitism for 1H at 29 °C by *Leptopilina boulardi* (G486 avirulent strain[110],). Larvae were allowed to develop further at 29 °C before dissection. Lymph glands were classified into three groups: disrupted when anterior lobes were absent or rudimentary, disrupting when some cells had escaped the anterior lobes, or intact when the anterior lobe border was regular. The percentage of lymph gland dispersal was calculated by scoring the number of lymph glands in each group, divided by the total number of infected larvae. Wasp egg encapsulation was analyzed 48H post parasitism and larvae were dissected. The number of fly larvae containing melanized/unhatched wasp eggs or living hatched wasp larvae, was counted. The % of wasp egg encapsulation was calculated by scoring the number of encapsulated wasp larvae inside the body of the dissected fly larvae, divided by the total number of infected fly larvae. In all experiments, genotypes were analyzed in parallel. Each experiment was repeated independently at least three times, and quantification represents the mean of at least three independent experiments. Graphs and statistical analyses using Pearson's Chi-squared test were performed using GraphPad Prism five software.

## Larval bleeding and hemocyte counting

Individual parasitized larvae were bled on microscope slides, and in parallel each larva was examined for lymph gland disruption. Only hemolymph from larvae with a bursted lymph gland was analyzed. Hemolymph samples were dried 10 min, fixed for 10 min in 4% paraformaldehyde in 1XPBS, then stained 2H with Phalloidin (1/100, Sigma). Samples were mounted in Vectashield (Vector Laboratories), examined using the Zeiss Celldiscoverer 7 and analyzed using the Zeiss Arivis software. For control (HandΔ>wt) and (HandΔ>5-HT1B-RNAi) at least 10 independent bleedings were analyzed per sample. The percentage of lamellocytes was calculated relative to the total number of circulating hemocytes. Lamellocytes are easily distinguishable from plasmatocytes by their elongated shape. Graphs and statistical analyses using Mann-Whitney nonparametric test, two-sided were performed using GraphPad Prism five software.

## Quantification of Trol and Laminin

Optimised z-stacks were made using the SP8 confocal microscope. Nuclei labelled by DAPI allowed us to measure the volume of each anterior lymph gland lobe. The volumes (in μm³) of Trol and Laminin staining were measured using Fiji software and 3DSuite plugin[111]. Trol and Laminin indexes correspond to (Trol or Laminin volume/anterior lobe volume) x10,000. At least 15 lymph gland anterior lobes were

scored per genotype. Graphs and statistical analyses using Mann-Whitney nonparametric test, two-sided were performed using Graph-Pad Prism five software.

## Quantification of 10xStat92E-GFP

Optimised z-stacks were made using the SP8 confocal microscope, and 3D reconstruction was performed using Imaris software. DAPI labelling was used to identify cardiac cell nuclei, which were defined as ROIs. The cardiac cell nuclei were cropped using Fiji (ImageJ) software. GFP and DAPI intensities for each ROI were quantified. The mean GFP intensity corresponds to the sum of GFP intensity/number of DAPI voxels per ROI. At least 40 nuclei from 10 lymph glands were scored per genotype, and experiments were reproduced at least four times. Graphs and statistical analyses using Mann-Whitney nonparametric test, two-sided were performed using GraphPad Prism five software.

## Zymography assays

Lymph glands were dissected at 4 °C in Schneider's medium, and incubated in 100μg/ml of DQ-gelatin conjugated fluorescein (Invitrogen) for 30 min in the dark. Lymph glands were rinsed 3 times with Schneider's medium, and then fixed in 4% EM-Grade paraformaldehyde for 25 min. They were mounted in Vectashield before imaging. Graphs and statistical analyses using Mann-Whitney nonparametric test, two-sided were performed using GraphPad Prism five software.

## Blood cell and progenitor quantification

Crystal cells were visualized by immunostaining with antibodies against Hnt. Plasmatocytes and core progenitors were labelled by P1 and Col immunostaining, respectively. Nuclei are labelled by DAPI. Number of crystal cells and volume (in μm³) of each anterior lymph gland lobe were measured using Fiji software and 3DSuite plugin[111]. Crystal cell index corresponds to [(number of crystal cells)/(anterior lobe volume)] x10,000. Plasmatocyte or progenitor indexes correspond to (plasmatocyte or progenitor volume/anterior lobe volume) x10,000. Lamellocyte or H3P indexes correspond to (lamellocyte or H3P volume/anterior lobe volume) x10,000. At least 15 anterior lobes per genotype were scored, and experiments were reproduced at least three times. Graphs and statistical analyses using t-test (Mann-Whitney nonparametric test, two-sided) were performed using GraphPad Prism five software.

## Heart rate measurements

Experiments were performed as described in ref. 46.

**Drosophila genetics.** Fly crosses for each figure:

Figure 1A, B"': *5-HT1B-Gal4(KI)* crossed with *UAS-LacZ^{nls}*; **1D and 1F:** *HandΔ-Gal4, UAS-dicer* crossed with *w^{1118}* (control) or *UAS-5-HT1B-RNAi* or *5-HT1B^{ΔIII-V}* homozygous mutant; **1E** :*76E11-Gal4, UAS-dicer* crossed with *w^{1118}* (control) or *UAS-5-HT1B-RNAi*.

Figure 2A: *tow-lexA* crossed with *w^{1118}* (control), **2B-B":** *Gyc89Da-Gal4* crossed with *UAS mcD8-GFP*. **2C-D:** *Gyc89Da-Gal4* crossed with *w^{1118}* (control) or *UAS-Trh-RNAi* or *UAS-Trh-RNAi2*.

Figure 3A: HandΔ-Gal4, UAS-dicer crossed with UAS-mCD8-GFP; **B-B":** HandΔ-Gal4; tow-LexA crossed with UAS-CD4-SpGFP1-10; LexAop-CD4-spGFP11; **C-D:** Gyc89Da-Gal4 crossed with w^{1118} (control) or UAS-CaMKII-RNAi; **E-F:** Gyc89Da-Gal4 crossed with w^{1118} (control) or UAS-Irk2.

Figure 4A–F: *HandΔ-Gal4, UAS-dicer* crossed with *w^{1118}* (control) or *UAS-5-HT1B-RNAi*; **4G-I:** *Gyc89Da-Gal4* crossed with *w^{1118}* (control) or *UAS-Trh-RNAi*; **4J-K:** *HandΔ-Gal4, UAS-dicer* crossed with *w^{1118}* (control) or *UAS-5-HT1B-RNAi*.

Figure 5A-B: HandΔ-Gal4, UAS-dicer crossed w^{1118} (control) or 5-HT1B-RNAi; **4D-E:** Gyc89Da-Gal4 crossed with w^{1118} (control) or UAS-Trh-RNAi; **4G-H:** HandΔ-Gal4, UAS-dicer crossed w^{1118} (control) or

5-HT1B-RNAi; **4J:** HandΔ-Gal4, UAS-dicer crossed with w[1118] (control) or UAS-5-HT1B-RNAi or UAS-Timp; UAS-5-HT1B-RNAi.

Figure 6**A:**10xStat92E-GFP; **B:** HandΔ-Gal4, UAS-dicer; 10x Stat92E-GFP crossed with w[1118]; **C:** HandΔ-Gal4, UAS-dicer;10xStat92E-GFP crossed with UAS-5-HT1B-RNAi; **E-F:** HandΔ-Gal4, UAS-dicer crossed with w[1118] (control) or hop-RNAi; **G-H:**

*HandΔ-Gal4, UAS-dicer crossed with UAS-5-HT1B-RNAi or hop-RNAi or UAS-hop^{Tum}; UAS-5-HT1B-RN*A;.**I-J***: HandΔ-Gal4, UAS-dicer crossed with hop-RNAi.*

Supplementary Fig. 1**A:** *5-HT1A-Gal4(KI)* crossed with *UAS-mCD8-GFP*; **B:** *5-HT2B-Gal4(KI)* crossed with *UAS-mCD8-GFP*; **C:** *5-HT7-Gal4(KI)* crossed with *UAS-GFP^{nls}*. **D:** *GMR64A03-Gal4* crossed with *UAS-mCD8-GFP*; **E:** *HandΔ-Gal4, UAS-dicer* crossed with *w[1118]* (control) or UAS-5-HT1B-*RNAi*; **F:** *76E11-Gal4, UAS-dicer* crossed with *w[1118]* (control) or *UAS-5-HT1B-RNAi2*; **G:** *GMR64A03-Gal4* crossed with *w[1118]* (control) or *UAS-5-HT1B-RNAi*;.**H:** *HandΔ-Gal4* crossed with *w[1118]* or *UAS-5-HT1B-RNAi2*; **I:** *HandΔ-Gal4 crossed with UAS-mCD8-GFP*; **J:** *HandΔ-Gal4, UAS-mCD8-GFP* crossed with *UAS-5-HT1B-RNAi*; **K:** *HandΔ-Gal4 crossed with* crossed *w[1118]* (control) or *5-HT1B-RNAi*; **L-T:** *HandΔ-Gal4; Gal80^{ts} crossed with* crossed *w[1118]* (control) or *5-HT1B-RNAi.*

Supplementary Fig. 2**A:** *GMR36B11-Gal4* crossed with *UAS-mCD8-GFP*; **B:** *GMR13C09-Gal4* crossed with *UAS-mCD8-GFP*; **C:** *Elav-Gal4(X)* crossed with *UAS-mCD8-GFP*; **D:** *Tow-LexA* crossed with *LexAop2-mCD8-GFP*; **E:** *Gyc89Da-Gal4* crossed with *w[1118]* (control) or *UAS-Trh-RNAi or UAS-Trh-RNAi2*; **F:** *GMR36B11-Gal4* crossed with *w[1118]* (control) or *UAS-Trh-RNAi or UAS-Trh-RNAi2*; **G:** *GMR13C09-Gal4* crossed with *w[1118]* (control) or *UAS-Trh-RNAi or UAS-Trh-RNAi2*; **H:** *elav-Gal4* (X) crossed with *w[1118]* (control) or *UAS-Trh-RNAi or UAS-Trh-RNAi2*; **I-Q:** *Gyc89Da-Gal4* crossed with *w[1118]* (control) or *UAS-Trh-RNAi.*

Supplementary Fig. 3**A:** *GMR36B11-Gal4* crossed with *w[1118]* (control) or *UAS-CaMKII-RNAi*; **B:** *GMR13C09-Gal4* crossed with *w[1118]* (control) or *UAS-CaMKII-RNAi*; **C:** *Elav-Gal4* (X) crossed with *w[1118]* (control) or *UAS-CaMKII-RNAi.*

Supplementary Fig. 4**A**–**H:** *HandΔ-Gal4, UAS-dicer* crossed with *w[1118]* (control) or *UAS-5-HT1B-RNAi*; **J-K:** *Gyc89Da-Gal4* crossed with *w[1118]* (control) or *UAS-Trh-RNAi*; **M-N:** *HandΔ-Gal4, UAS-dicer* crossed with *w[1118]* (control) or *UAS-5-HT1B-RNAi*; **O:** *HandΔ-Gal4, UAS-dicer* crossed with *w[1118]* (control) or *UAS-Timp*; **P-R:** *HandΔ-Gal4, UAS-dicer* crossed with *UAS-5-HT1B-RNAi or UAS-Timp; UAS-5-HT1B-RNAi.*

Supplementary Fig. 5A, B'**:** *10xStat92E-GFP*; **C:** *HandΔ-Gal4, UAS-dicer* crossed with *w[1118]* (control) or *UAS-hop^{Tum}*; **D-F:** *HandΔ-Gal4, UAS-dicer* crossed with *UAS-5-HT1B-RNAi or UAS-hop^{Tum}; UAS-5-HT1B-RNAi.*

### Reporting summary

Further information on research design is available in the Nature Portfolio Reporting Summary linked to this article.

## Data availability

The authors declare that the data supporting the findings of this study are available within the article and its Extended Data Figs. Source data are provided with this paper.

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

## Acknowledgements

We thank: M. Delatre, N. Formicola, J. Hombria, S. Kondo, F. Pinto Teixeira, Bloomington and Vienna Stock Center, and the TRiP at Harvard Medical School for fly strains; I. Ando, S. Baumgartner, A. Moore and T. Trenczek for antibodies, V. Gobert, M. Haenlin, G. Lebreton, M. Meister, C. Monod for critical reading of the manuscript. We are grateful to B. Ronsin and S. Bosch for assistance with confocal microscopy (Platform TRI), J. Favier, V. Nicolas and A. Destenable for fly culture; D. Jullien, V. Dougados and C. Chailleux for technical assistance, and A. Paululat and

B. Guiard for their advice during the course of X. Liu's thesis. Research in the authors' laboratory is supported by the CNRS, University Toulouse III, FRM (Fondation pour la Recherche Médicale) DEQ20180339171 (MC), La Société Française d'Hématologie (SFH for X.L), the China Scholarship Council (X.L), the CNRS « Groupement de recherche 3740 » and ANR grant (Number: 279956).

## Author contributions

Conceived and designed the experiments: X.L., N.V. and M.C.; performed the experiments: X.L., M.M. and N.V.; analyzed the data: X.L., M.M., N.V. and M.C.; and wrote the paper: X.L. and M.C.

## Competing interests

The authors declare no competing interests.
