## [Peer Review file · Nature Communications]

Serotonergic neurons regulate the *Drosophila* vascular niche to control immune stress hematopoiesis

Corresponding Author: Dr Michele Crozatier

Version 0:

Reviewer comments:

Reviewer #1

(Remarks to the Author)

This is a lovely paper, which demonstrates for the first time that serotonin signaling mediates a neural influence on hematopoiesis in *Drosophila* larvae, via a vascular 'niche' in the anterior aorta. The major conclusions are well supported in several ways by nicely carried out experiments. I have only a few points of criticism to the text:

1. In general, I object to the excessive use of abbreviations. They make reading unnecessarily slow and boring, even for a person who is acquainted to the field (and even worse for someone who is not). For instance, do we really need abbreviations like 'CT' for the cardiac tube, or 'LG' for the lymph gland? The space saved by these abbreviations is insignificant, but the lost focus of the reader is not. Similarly, I'm not sure if non-drosophilists know that 'the L3 stage' is jargon for third instar larvae.
2. I was surprised to read that 'the contribution of vascular niche cells has not been described so far' (line 63), given that the contribution of vascular niche cells has been nicely documented by the Crozatier lab (Ref. 15).
3. On line 85, should it be 'phagocytosis BY insect hemocytes'?
4. On lines 102-104 it is mentioned that 'Single-cell RNA-sequencing data from our lab indicate that the serotonin receptor 1B (5-HT1B) is expressed in larval aorta cells (Morin-Poulard et al., in preparation)'. However, published single-cell RNA-sequencing data also show that 5-HT1B is expressed in the posterior signalling center, and in a class of circulating blood cells ('primocytes'), which are similar or identical to the cells in that center, as reported by Cattenoz et al. 2020 (doi 10.15252/embj.2020104486), Cho et al. 2020 (doi 10.1038/s41467-020-18135-y) and Girard et al. 2021 (doi 10.7554/eLife.67516), and summarised by Hultmark & Andó 2022 (doi 10.7554/eLife.78906). Furthermore, Rodrigues et al 2021 (doi 10.1101/2021.06.14.44909) report that a 5-HT1B reporter is expressed in the posterior lobes and in the lymph gland medulla, but apparently not in the aorta. These observations may affect some of the conclusions and they should be discussed.
5. Please give a reference for the tissue specificity of the Hand Δ -Gal4 driver (again, presumably Ref. 15).
6. On lines 129-148, the effect of suppressing 5-HT1B expression in the aorta is described. But, what if 5-HT1B is also expressed in the posterior signaling center, as suggested in point 4 above? Is it possible that the 5-HT1B-Gal4 reporter does not fully reflect the endogenous 5-HT1B gene?
7. For the 'inward-rectifier potassium channel' gene, please use the FlyBase nomenclature: *Irk2*, not *Kir2.1*.
8. The term 'stress hematopoiesis' (line 210 and elsewhere) is not well chosen. The response described in this manuscript is better described as an immune response, and sometimes also a wound response. 'Stress response' is a very vague term that often refers to the heat shock response, and to the responses to various chemicals or other insults, as well as to the release of stress hormones and effects on the nervous system.
9. On lines 214-216 it is stated that 'very few lamellocytes derived from the trans-differentiation of sessile/circulating hemocytes are detected in the hemolymph prior to LG dispersal'. Now, the relative contribution of lymph gland versus peripheral hematopoietic sites to the blood cells (and specifically lamellocytes) to the immune response can be debated. For

instance, Honti et al. found that in their experiments the lymph gland contributed only about 8% of the circulating lamellocytes. My feeling is that this difference could depend on the timing of lymph gland disruption, which may happen 10 h after infection as reported in this manuscript, or as late as 75 h as reported by Sorrentino et al. 2002 (doi 10.1006/dbio.2001.0542). These differences may depend on the fly strain and the exact timing of infection, or on the definition of lymph gland dispersal (first detection of basal membrane degradation, versus complete release of the contents).

10. In the Discussion (and perhaps in the Introduction) it would be appropriate to refer also to the work from Katja Brückner's lab about the control of peripheral hematopoietic sites by the peripheral nervous system.

Figure 3. What are the white arrowheads in panel A?

Reviewer #2

(Remarks to the Author)

In this manuscript, Crozatier and coauthors have identified an interesting function of serotonin receptors in the aorta in the lymph gland dispersal phenotype upon wasp parasitism. There are a few main concerns to be addressed to substantiate the authors' claims.

Main points>

1. Although the wasp egg encapsulation rate and the lymph gland disruption may be causally related, the most significant impact influencing wasp egg encapsulation following the lymph gland disruption is the change in circulating lamellocyte numbers. The authors visualized lymph gland phenotypes, including proliferation, differentiation, and ECM expression at 6 to 8H post parasitism. They also assessed the ratio of circulating lamellocytes in 5HT1B RNAi larvae upon wasp parasitism. However, it remains unclear whether the minor reduction in circulating lamellocytes caused by 5HT1B RNAi, as shown in 4J, is sufficient to inhibit the wasp egg encapsulation as seen in 1D-E. Additionally, it is uncertain whether this reduction in the number of circulating lamellocytes is observed in other genotypes, including those with timp overexpression-mediated encapsulation rescue. A direct causal link between lymph gland integrity and reduced wasp egg encapsulation and changes in circulating lamellocytes requires further substantiation.

2. In Figures 2c-d, 3c-f, 4g-l, 5d-f, the authors used Gyc89da-gal4 to downregulate the function of aorta neurons involved in the lymph gland disintegration. However, given that Gyc89da-gal4 is broadly expressed in various sensory neurons for ambient oxygen sensation in the brain (Morton et al., J. Exp. Biol 2008), its use as a Gal4 for aorta-specific neurons may not adequately address the specificity required for the authors' claims. This concern also applies to other gal4 lines, GMR36B11 and GMR13C09. To address this limitation, the authors could either identify upstream neurons of the aorta neurons expressing Gyc89da-gal4 or use a combination of Gyc89da-gal4 with another aorta neuron specific driver through gal4 AD-DBD system to validate their observations.

3. What is the known function of JAK/STAT pathway in the aorta? The function of JAK/STAT pathway in blood progenitors is well understood. However, its role in the aorta remains unclear. Is JAK/STAT the only pathway controlled by the serotonin receptor or are other pathways involved? Furthermore, is JAK/STAT downregulation under homeostatic conditions sufficient to induce timp expression or reduce ECM expression in the lymph gland? It is unclear whether the Hand>hop_TumL;5HT1B RNAi mediated rescue of wasp encapsulation and lymph gland disruption directly influences the ECM degradation. The specificity and directness of the serotonin-JAK/STAT pathway in ECM control need to be substantiated to fully support the authors' claims.

4. Throughout the manuscript, the authors primarily presented their data using dynamite plots. For example, Fig 1D-F, Fig 2C-D, Fig 3C-F, Fig 4J, Fig 5J, Fig 6E-H, Sup Fig 1E-G and H, Sup Fig 2E-H, Sup Fig. 3A-C, Sup Fig 4O, and Sup Fig 5C. Given the variability often observed in lymph gland phenotypes, it is strongly recommended to adopt data visualization formats such as box-whisker plots with individual data points. This approach would better illustrate the range of variability and sample sizes and provide a clear representation of the data.

Minor points>

1. Although single-cell RNA seq data is now in preparation, the specific mRNA expression of 5-HT1B in anterior aorta cells needs to be validated ideally using qRT-PCR or in situ hybridization. In line with this comment, RNAi efficiencies against 5HT-1B require validation.

2. The manuscript discusses neuron-immune interactions in the context of mechanical nociceptors and wasp parasitism. However, the possible regulatory mechanisms of serotonin release under normal development as well as in wasp infestation remain unclear. Discussing this point in the discussion would provide valuable insights into serotonin involvement and its physiological relevance to this process.

Version 1:

Reviewer comments:

Reviewer #1

(Remarks to the Author)

In the revised version of this manuscript, together with the rebuttal letter, the authors have dealt with the questions raised by

me and the other referee in a fully satisfactory way.

I note, however, that something has gone wrong in the list of references, specifically with the doi links. This must be fixed.

Reviewer #2

(Remarks to the Author)

The authors have thoroughly addressed my concerns and have provided additional experiments to support the mechanistic details. I strongly recommend this manuscript for publication in Nature Communications.

REVIEWER COMMENTS

Reviewer #1 (Remarks to the Author):

This is a lovely paper, which demonstrates for the first time that serotonin signaling mediates a neural influence on hematopoiesis in *Drosophila* larvae, via a vascular 'niche' in the anterior aorta. The major conclusions are well supported in several ways by nicely carried out experiments. I have only a few points of criticism to the text:

We would like to thank the reviewer for her/his kind comments, and we appreciate her/his interest in our article as well as her/his useful comments on how to improve it.

1. In general, I object to the excessive use of abbreviations. They make reading unnecessarily slow and boring, even for a person who is acquainted to the field (and even worse for someone who is not). For instance, do we really need abbreviations like 'CT' for the cardiac tube, or 'LG' for the lymph gland? The space saved by these abbreviations is insignificant, but the lost focus of the reader is not. Similarly, I'm not sure if non-drosophilists know that 'the L3 stage' is jargon for third instar larvae.

We apologize for any inconvenience caused by the use of abbreviations. We have now modified the main text accordingly and written lymph gland and cardiac tube in full. However, for fly genotypes using the cardiac tube (CT) driver, we have used CT in order to keep the name shorter. We have also written "second or third instar larvae" instead of "L2 or L3 stage".

2. I was surprised to read that 'the contribution of vascular niche cells has not been described so far' (line 63), given that the contribution of vascular niche cells has been nicely documented by the Crozatier lab (Ref. 15)

Sorry for the misunderstanding. In line 63 we talk about "the role of the vascular niche in response to parasitism" and not under physiological conditions. For the sake of clarity, we rewrote the sentence (lines 64-65).

3. On line 85, should it be 'phagocytosis BY insect hemocytes'?

Yes, this has been modified.

4. On lines 102-104 it is mentioned that 'Single-cell RNA-sequencing data from our lab indicate that the serotonin receptor 1B (5-HT1B) is expressed in larval aorta cells (Morin-Poulard et al., in preparation)'. However, published single-cell RNA-sequencing data also show that 5-HT1B is expressed in the posterior signalling center, and in a class of circulating blood cells ('primocytes'), which are similar or identical to the cells in that center, as reported by Cattenoz et al. 2020 (doi 10.15252/embj.2020104486), Cho et al. 2020 (doi 10.1038/s41467-020-18135-y) and Girard et al. 2021 (doi 10.7554/eLife.67516), and summarised by Hultmark & Andó 2022 (doi 10.7554/eLife.78906). Furthermore, Rodrigues et al 2021 (doi 10, e61409) report that a 5-HT1B reporter is expressed in the posterior lobes and in the lymph gland medulla, but apparently not in the aorta. These observations may affect some of the conclusions and they should be discussed.

Thank you for your comment. In the previous version of the text, we forgot to mention the published data concerning the expression pattern of 5-HT1B in blood cells. This information and the corresponding references have now been added in the discussion section. In this study, we further show that *5-HT1B* is expressed in anterior aorta cells. By knocking down *5-HT1B* in cardiac cells (using *HandΔ* or *76E11* or *GMR64A03*, Fig. 1 and Sup Fig. 1), we establish a cell-autonomous role for *5HT1B* in aorta cells to regulate lymph gland hematopoiesis in response to immune stress. The two drivers *HandΔ* and *76E11* are expressed in cardiac cells but not in tissues where *5HT1B* is expressed (Tian et al., 2023). Since *5-HT1B* is expressed in other cell types (see above), the question about its putative role in these cells requires further investigation through the use of appropriate drivers

The text in the revised version of the ms has been modified (see lines 349-357).

5. Please give a reference for the tissue specificity of the *HandΔ*-Gal4 driver (again, presumably Ref. 15).

This has been done. We added reference 15 (line 125).

6. On lines 129-148, the effect of suppressing 5-HT1B expression in the aorta is described. But, what if 5-HT1B is also expressed in the posterior signaling center, as suggested in point 4 above?

This is an interesting question which has yet to be addressed, because in this study, we have focused on the role of *5-HT1B* in cardiac cells. The two drivers, *HandΔ* and *76E11* that we used to KD *5-HT1B* are expressed in cardiac cells but not in the PSC (Tian et al., 2023).

Is it possible that the 5-HT1B-Gal4 reporter does not fully reflect the endogenous 5-HT1B gene?

To analyze the expression of *5-HT1B*, we used two transgenic knock-in lines that were generated independently by Kondo et al., 2020 and Deng et al., 2020. In both lines, we observed expression in the cells of the larval anterior aorta, but we never detected expression in the MZ or cells of the posterior lobes, suggesting that indeed the driver only partially reproduces receptor expression. However, in approximately 10% of lymph glands, we observed that one or two cells per PSC were labelled, which is in agreement with published scRNAseq data (Cho et al., 2020).

7. For the 'inward-rectifier potassium channel' gene, please use the FlyBase nomenclature: *Irk2*, not *Kir2.1*.

This has been modified in the text (lines 214) and in Fig. 3E-F.

8. The term 'stress hematopoiesis' (line 210 and elsewhere) is not well chosen. The response described in this manuscript is better described as an immune response, and sometimes also a wound response. 'Stress response' is a very vague term that often refers to the heat shock response, and to the responses to various chemicals or other insults, as well as to the release of stress hormones and effects on the nervous system.

To take into account the reviewer's comment, we changed the title and the text and wrote "immune stress hematopoiesis" instead of "stress hematopoiesis".

9. On lines 214-216 it is stated that 'very few lamellocytes derived from the trans-differentiation of sessile/circulating hemocytes are detected in the hemolymph prior to LG dispersal'. Now, the relative contribution of lymph gland versus peripheral hematopoietic sites to the blood cells (and specifically lamellocytes) to the immune response can be debated. For instance, Honti et al. found that in their experiments the lymph gland contributed only about 8% of the circulating lamellocytes. My feeling is that this difference could depend on the timing of lymph gland disruption, which may happen 10 h after infection as reported in this manuscript, or as late as 75 h as reported by Sorrentino et al. 2002 (doi 10.1006/dbio.2001.0542). These differences may depend on the fly strain and the exact timing of infection, or on the definition of lymph gland dispersal (first detection of basal membrane degradation, versus complete release of the contents).

In our opinion, comparison with various published data is not really relevant, because the conditions used in each study are not equivalent. In some studies, the wasps were allowed to infect for up to 24 hours (1 hour in our study), and the temperature at which the larvae were reared varied from 18°C to 29°C (29°C in our study). Furthermore, in most of the studies, the encapsulation of wasp eggs was not assessed, so no correlation can be established between the number of circulating lamellocytes and the effect on wasp egg neutralization. In conclusion, similar experimental conditions are essential to allow comparison of results.

With regard to the contribution of the fly genotype to the time of rupture of the lymph glands in response to parasitism, we observed a slight difference depending on the Gal4 drivers used, with a maximal variation of around 2 hours. Approximately 50% of lymph glands are intact 10 and 12 hours after parasitism in *HandΔ>w* (Fig. 1) and *Gyc-Gal4>w* (Fig. 3), respectively. To avoid this problem, it is essential to compare controls and assays using the same driver background.

10. In the Discussion (and perhaps in the Introduction) it would be appropriate to refer also to the work from Katja Brückner's lab about the control of peripheral hematopoietic sites by the peripheral nervous system.

In the previous version of the ms, data from Katja Brückner's laboratory were only briefly mentioned (see lines 365-366 in the previous version of the paper). To address the reviewer's comment, we now provide more detailed information on the control of sessile hemocytes in hematopoietic pockets by neurons, and we cite the two corresponding articles (see lines 400-407). We refer to these data in the discussion and not in the introduction, because they are not directly related to neuronal regulation of the lymph gland, which is specifically addressed in this study.

Figure 3. What are the white arrowheads in panel A?

The white arrowheads indicate the cardiac nuclei. This information has now been added to the figure legend.

Reviewer #2 (Remarks to the Author):

In this manuscript, Crozatier and coauthors have identified an interesting function of serotonin receptors in the aorta in the lymph gland dispersal phenotype upon wasp parasitism. There are a few main concerns to be addressed to substantiate the authors' claims.

We would like to thank the reviewer for her/his useful and valuable comments on how to improve the article. We also appreciate her/his interest in our ms.

Main points>

1. Although the wasp egg encapsulation rate and the lymph gland disruption may be causally related, the most significant impact influencing wasp egg encapsulation following the lymph gland disruption is the change in circulating lamellocyte numbers. The authors visualized lymph gland phenotypes, including proliferation, differentiation, and ECM expression at 6 to 8H post parasitism. They also assessed the ratio of circulating lamellocytes in *5HT1B-RNAi* larvae upon wasp parasitism.

Two independent studies established that lamellocytes derived from lymph glands are essential for the successful encapsulation of wasp eggs, and that the rupture of lymph glands and the release of lamellocytes into the circulation strongly correlate with the success of wasp egg encapsulation (Sorrentino et al., 2002; Louradour et al., 2017).

In addition, very few lamellocytes resulting from the trans-differentiation of sessile/circulating hemocytes are detected in the hemolymph prior to lymph gland dispersal, compared with the total number of circulating lamellocytes detected when the lymph gland disperses. This observation holds for all the genetic backgrounds we have tested to date (Louradour et al., 2017 and our unpublished data), indicating that the lymph gland routinely loads substantial numbers of lamellocytes into the hemolymph. Accordingly, it was previously reported that a strong correlation exists between the amount of lamellocytes in the hemolymph and the ability of *Drosophila* larvae to efficiently encapsulate the wasp egg (Prevost et Eslin, 1998).

However, it remains unclear whether the minor reduction in circulating lamellocytes caused by *5HT1B-RNAi*, as shown in 4J, is sufficient to inhibit the wasp egg encapsulation as seen in 1D-E..

This reduction was monitored at the time of lymph gland dispersal in control larvae and in *5-HT1B-KD* larvae, which occurs at a different time since the *5-HT1B-KD* lymph gland ruptures prematurely (12 hours after parasitism instead of 14 hours in controls). To further address the potential impact of this reduction on wasp neutralization efficiency, we also quantified the percentage of circulating lamellocytes 21 hours post-parasitism, about the time wasp larvae emerge inside the body of the *Drosophila* larva (new Fig. 4K). The percentage of circulating lamellocytes is around 3-fold higher in the control than in *5-HT1B-KD*. These results indicate that the deficiency of circulating lamellocytes in *5-HT1B-KD* larvae is observed throughout wasp egg development and challenges its successful encapsulation. These novel data are added in Fig. 4K and in the text lines 246-252.

Additionally, it is uncertain whether this reduction in the number of circulating lamellocytes is observed in other genotypes, including those with timp overexpression-mediated encapsulation rescue. A direct causal link between lymph gland integrity and reduced wasp egg encapsulation and changes in circulating lamellocytes requires further substantiation.

In response to the reviewer's comments and to reinforce the correlation between lymph gland rupture, the percentage of circulating lamellocytes and wasp egg encapsulation capacity, we also measured the percentage of circulating lamellocytes in

HandΔ>hop^{Tum}; 5-HT1B-RNAi (see Figure below) where a significant proportion of lymph glands disperse later (Fig. 6H) and encapsulate better (Fig. 6G) compared to *5-HT1B-KD* alone. We also observed a slight increase in the percentage of circulating lamellocytes in *HandΔ>hop^{Tum}; 5-HT1B-RNAi* compared to *5-HT1B-RNAi*. However, the difference is not statistically significant, mainly due to the huge variation in the percentage of circulating lamellocytes from one larva to another, particularly in the controls, reflecting a disparity in their response to parasitism (see also Fig. 4J-K).

2. In Figures 2c-d, 3c-f, 4g-l, 5d-f, the authors used *Gyc89da-gal4* to downregulate the function of aorta neurons involved in the lymph gland disintegration. However, given that *Gyc89da-gal4* is broadly expressed in various sensory neurons for ambient oxygen sensation in the brain (Morton et al., J. Exp. Biol 2008), its use as a Gal4 for aorta-specific neurons may not adequately address the specificity required for the authors' claims. This concern also applies to other gal4 lines, *GMR36B11* and *GMR13C09*. To address this limitation, the authors could either identify upstream neurons of the aorta neurons expressing *Gyc89da-gal4* or use a combination of

Gyc89da-gal4 with another aorta neuron specific driver through gal4 AD-DBD system to validate their observations.

To inhibit serotonin synthesis, we used four independent neuronal drivers. They are all expressed in brain neurons and share a common expression in Neurons^{aorta}. Since the same hematopoietic effect was observed with all four drivers, it is reasonable to propose that this shared effect was due to serotonin reduction in Neurons^{aorta}. Using pairs of Neurons^{aorta} drivers combined with the Gal4 AD-DBD system we could restrict the number of candidate serotonergic neurons that are involved here, but this would still be insufficient to firmly identify the neurons producing serotonin to activate 5-HT1B in aorta cells. The next steps required are the identification of which among the hundreds of serotonergic neurons present in the larval brain activate serotonin signaling in the aorta, and the generation of drivers to inhibit specifically their function. This is an entirely independent study that could take years and which we would like to address in the near future, but it remains outside of the scope of this manuscript.

To address the reviewer's concern and to be more cautious about the conclusions drawn from the data, we now conclude that serotonin produced by neurons (instead of neurons innervating the aorta, therein called Neurons^{aorta}) controls serotonin signaling in aorta cells. In the results and discussion sections, we only suggest that serotonin released by Neurons^{aorta} may be involved. We have accordingly modified the article title, summary and Fig. 7, as well as rewritten part of the text (see lines 184-187; 430-440).

3. What is the known function of JAK/STAT pathway in the aorta? The function of JAK/STAT pathway in blood progenitors is well understood. However, its role in the aorta remains unclear.

To our knowledge there are no data in the literature about a role of the JAK/STAT pathway in larval aorta cells to control lymph gland hematopoiesis. Our study is the first to establish that this pathway is activated in aorta cells in response to parasitism. Furthermore, it shows that JAK/STAT activation in aorta cells in response to parasitism is required to prevent lymph gland rupture and allows efficient wasp egg encapsulation (Fig. 6). In addition, we demonstrate that JAK/STAT activation in aorta cells controls the lymph gland Trol meshwork in response to parasitism but not underhomeostatic conditions (see novel data presented below). Collectively these observations indicate

that JAK/STAT signaling, as does serotonin signaling, regulates the extracellular matrix of the lymph gland in response to parasitism. Studying the autonomous function of JAK/STAT signaling in aorta cells will require further investigation to identify the intracellular signaling molecules involved downstream of this pathway and required to control of lymph gland hematopoiesis by 5-HT1B.

Is JAK/STAT the only pathway controlled by the serotonin receptor or are other pathways involved?

The identification of other 5-HT1B-dependent signaling pathways regulating lymph gland hematopoiesis requires further investigation.

Furthermore, is JAK/STAT downregulation under homeostatic conditions sufficient to induce *timp* expression or reduce ECM expression in the lymph gland?

Since the reporter *10XStat92E-GFP* is barely detected in aorta cells under homeostatic conditions (Fig. 6A), our interpretation is that this pathway is not activated in these conditions. However, to address the reviewer's concern, we functionally tested whether the JAK/STAT pathway is involved in lymph gland extracellular matrix organization in the absence of parasitism. To do so, we knocked down *hop* in cardiac cells (*CT>hop-RNAi*) and looked at Trol expression. There was no difference compared to control. Please see the results on the Figure provided below. This result further supports that JAK/STAT signaling is not required in cardiac cells under homeostatic conditions to regulate lymph gland extracellular matrix.

Concerning *timp*: we do not have any indication that *timp* is involved in aorta cells to regulate lymph gland extracellular matrix. In our study we only used Timp as a tool to inhibit metalloproteinase activity. In the revised version we analyzed Trol staining in 5-

HT1B KD and *CT>Timp>5HT1B-RNAi*. These novel data are given in Sup Fig. 4P-R. A slightly higher Trol staining is observed in *CT>Timp>5HT1B-RNAi* compared to *5-HT1B-RNAi* alone. This is an agreement with the slight delay in lymph gland rupture observed (Fig. 5J).

It is unclear whether the *Hand>hop_TumL;5HT1B RNAi* mediated rescue of wasp encapsulation and lymph gland disruption directly influences the ECM degradation. The specificity and directness of the serotonin-JAK/STAT pathway in ECM control need to be substantiated to fully support the authors' claims

To address the reviewer's comment, we analyzed Trol expression in response to parasitism when the JAK/STAT pathway was inhibited in cardiac cells (*CT>hop-RNAi*). In 6 hours post-parasitism *CT>hop-RNAi*, lymph glands display a significant decrease in Trol level as compared to the control (new Fig. 6I-K), indicating that in response to parasitism JAK/STAT signaling activated in cardiac cells regulates the lymph gland extracellular matrix. We then tested Trol expression when *hop^{Tum}* is expressed in *5-HT1B-KD* (*CT>hop^{Tum}; 5-HT1B-RNAi*) (new data in Sup. Fig. 5D-F). A significant rescue of Trol expression was observed compared to *CT>5-HT1B-KD* alone. This is in agreement with the delay in lymph gland rupture (Fig. 6H) and increased wasp egg encapsulation (Fig. 6G) observed in this context. Taken together, these data suggest that in response to parasitism, 5-HT1B and JAK/STAT signaling are involved in the same process, and that in aorta cells 5-HT1B functions, at least in part through JAK/STAT signaling, to prevent lymph gland extracellular matrix degradation.

4. Throughout the manuscript, the authors primarily presented their data using dynamite plots. For example, Fig 1D-F, Fig 2C-D, Fig 3C-F, Fig 4J, Fig 5J, Fig 6E-H, Sup Fig 1E-G and H, Sup Fig 2E-H, Sup Fig. 3A-C, Sup Fig 4O, and Sup Fig 5C. Given the variability often observed in lymph gland phenotypes, it is strongly recommended to adopt data visualization formats such as box-whisker plots with individual data points. This approach would better illustrate the range of variability and sample sizes and provide a clear representation of the data.

For wasp egg encapsulation tests, the output is binary and cannot be represented as quantitative data point for each individual larva: the wasp egg is encapsulated or not.

Thus, we monitored encapsulation by measuring the percentage of wasp eggs that are encapsulated. The total number of parasitized larvae analyzed is indicated on the histograms.

Similar analyses are carried out for lymph gland disruption, but in this case, there are three possibilities: the lymph glands are intact, disrupted or disrupting.

We think that data visualization using box-whisker plots, as suggested by the reviewer, is not suitable for this type of experiment. We believe that dynamite plots are the simplest way to represent the results, which is in line with the representations previously used in the literature. In order to clarify these data representations for the reader, we now explain how the quantifications were carried out in the legend to Figure 1, in addition to the information provided in Materials and Methods.

Following the reviewer's advice, we have changed the way we represent the percentage of circulating lamellocytes (Fig. 4J-K). Box-whisker plots better illustrate the variability between larvae.

Minor points>

1. Although single-cell RNA seq data is now in preparation, the specific mRNA expression of 5-HT1B in anterior aorta cells needs to be validated ideally using qRT-PCR or in situ hybridization. In line with this comment, RNAi efficiencies against 5HT-1B require validation.

Thank you for your comment. Indeed, in the previous version of the ms we forgot to mention the published data concerning the expression pattern of 5-HT1B in blood cells. We now added a paragraph in the discussion (lines 349-357) to summarize these data. In our study, we further established, by using two knock-in transgenic lines, that 5-HT1B is expressed in larval anterior aorta cells.

We generated a *5-HT1B* fluorescent RNA probe using Stellaris technology and performed *in situ* hybridization (*ISH*). The results are shown in the figure below. We detected *5-HT1B* transcripts in neurons of the larval brain (A-A") and in ring gland cells (B-B'), indicating that the probe is fine. However, whereas in most lymph glands (30 lymph glands analyzed) no staining was observed, sporadic labelling in a few lymph glands and aorta cells was observed. In 7 out of 30 lymph glands, 1 to 3 aorta cells were labelled (C-C'). In other lymph glands (in 2 out of 60 anterior lobes), 1 or 2 PSC cells expressed *5-HT1B* (D-D'), and in other lymph glands (12 out of 60 anterior lobes)

we detected 1 to 2 cells in the lymph gland itself (E-E'). The sporadic and erratic detection of *5-HT1B* transcripts in lymph gland and aorta cells suggests that *5-HT1B* is expressed at very low levels, at the limit of detection using this approach. Even though not all lymph glands are labelled, these data are in agreement with data from the literature, establishing expression in progenitors and PSC cells by using different reporter lines and single cell RNAseq. Finally, these data confirm that *5-HT1B* is expressed in aorta cells, as revealed by the two *5-HT1B* *KI* lines used in this study and our unpublished single cell RNA seq data.

Figure legend

(A-E) ISH of 5-HT1B (red or white) in col >mcd8GFP (green) larvae. (A-A'') larval brain, (B-B') ring gland, (C-E'') lymph glands are shown, (C, D-D', E) the PSC is in green. (A-A'') 5-HT1B (red in A-A' and white in A'') is expressed in neurons (arrow head). (B-B') a subset of ring gland cells expresses 5-HT1B, red in B and white in B'. (C-C') three aorta cells express 5-HT1B (white and arrow heads). (D-D') 5-HT1B (white) is expressed in two PSC cells (green, arrows) and in one aorta cell (arrow head). (E-E') 5-HT1B (white) is expressed in one lymph gland cell (red).

Brain

Ring gland

Lymph gland

The two *5-HT1B-RNAi* lines that we used (25833 and 33418 from the Bloomington Stock Center) to knock down *5-HT1B* in cardiac cells were previously used to inhibit *5-HT1B* function in many different cell types (Li et al., *Elife* 2023; Qi et al., *Elife* 2016; Kaneko et al., *Neuron* 2017; Senapati et al., *Nat. Neurosci* 2019; Tsao et al., *Elife* 2018). Furthermore, for the 33418 RNAi line the efficiency of the RNAi treatment was validated by RT-qPCR in a previous study (Cao et al., *European Journal of Medical Research* 2022).

2. The manuscript discusses neuron-immune interactions in the context of mechanical nociceptors and wasp parasitism. However, the possible regulatory mechanisms of serotonin release under normal development as well as in wasp infestation remain unclear. Discussing this point in the discussion would provide valuable insights into serotonin involvement and its physiological relevance to this process.

We apologize for not being clear enough in the previous version of the ms. We found that serotonin produced by neurons and *5-HT1B* expressed by anterior aorta cells are not required to regulate lymph gland hematopoiesis under physiological conditions (no parasitism). The simplest explanation is that under these conditions, serotonin is not released by neurons. In contrast, in response to wasp parasitism, serotonin signaling is required to regulate lymph gland hematopoiesis to deal with this immune stress. Our current model is that wasp parasitism activates serotonin neurons, and in turn leads to the release of serotonin. Serotonin binds to the *5-HT1B* receptor expressed by anterior aorta cells and serotonin signaling activates, at least in part, the JAK/STAT pathway in a cell-autonomous manner to prevent lymph gland rupture. In the revised version of the ms we modified the text in order to clarify these points (see lines 407-414).